# Structural basis for uracil removal from DNA by human SMUG1

Julian M. Ludäscher [1], Emma Scaletti Hutchinson[1], Guillem Vila-Julià [2], Ann-Sofie Jemth [3], Saher Shahid[1], Elisee Wiita[3], Israel Cabeza de Vaca [2], Szymon Pach[2], Lukas Gajdos [4], Swati Aggarwal[5,6], Ellen Walse[1], Oliver Mortusewicz [3], Thomas Helleday [3], Jens Carlsson [2] & Pål Stenmark [1] ✉

Human single-strand-selective monofunctional uracil DNA glycosylase 1 (hSMUG1) removes uracil, 5-hydroxymethyluracil (5hmU) and 5-fluorouracil (5FU) from DNA, thereby initiating the base excision repair (BER) process. hSMUG1 is important for maintaining genomic integrity and plays a significant role in cancer biology. Here, we present the structures of hSMUG1, including complexes with products (uracil and 5FU) and an enzyme-product complex of hSMUG1 with double-stranded DNA (dsDNA). Analysis of our hSMUG1-dsDNA complex reveals how uracil is flipped out of the dsDNA for excision and identifies key residues that we confirm to be critical for both DNA binding and enzymatic activity. Furthermore, our hSMUG1 substrate complexes, molecular dynamics simulations and neutron diffraction data suggest a mechanism by which the substrate uracil rotates following base excision. The structural and functional information presented here will be highly useful for the future development of inhibitors and/or activators targeting hSMUG1.

The DNA of a cell is susceptible to damage from exogenous factors such as chemicals and radiation, or endogenous reactions including deamination, oxidation, methylation or even spontaneous decay of the DNA itself[1,2]. As this damage can lead to mutations, all organisms have evolved diverse controls and complex networks of DNA repair[3]. The base excision repair pathway (BER), discovered in *E. coli* by Nobel laureate Thomas Lindahl in 1974, is the predominant pathway for repairing small base lesions in DNA[4]. There are two different forms of BER: short-patch BER (SP-BER), in which only one faulty base is replaced, or long-patch BER (LP-BER), in which a longer DNA sequence is newly synthesised[5] (Fig. 1). Both processes share the same five essential steps: (1) base excision, (2) strand incision, (3) processing of the strand ends, (4) re-synthesis, and (5) ligation[6]. However, SP-BER and LP-BER use different enzymes to carry out these functions[7].

DNA glycosylases initiate BER by scanning the DNA and then excising damaged bases, which generates abasic (AP) sites[8]. These important enzymes are divided into six superfamilies based on their structural properties[9], and three classes based on their catalytic properties, namely, monofunctional, bifunctional and Nei-like glycosylases[10]. Monofunctional glycosylases only carry out base excision, which occurs via hydrolysis of the *N*-glycosidic bond between the DNA base and ribose, after which the AP site is processed further by enzymes with endonuclease activity, such as Apurinic/apyrimidinic Endonuclease 1 (APE1)[11]. Bifunctional glycosylases, on the other hand, have an additional AP lyase activity which allows them to also cleave the phosphate backbone by the mechanism of β-elimination after they have removed the damaged base[12]. Nei-like glycosylases are often counted among the bifunctional glycosylases as they also have AP-lyase activity. However, the mechanism differs slightly from

[1]Department of Biochemistry and Biophysics, Stockholm University, Stockholm, Sweden. [2]Science for Life Laboratory, Department of Cell and Molecular Biology, Uppsala University, Uppsala, Sweden. [3]Science for Life Laboratory, Department of Oncology-Pathology, Karolinska Institute, Stockholm, Sweden. [4]Large Scale Structures group, Institut Laue-Langevin (ILL), Grenoble, France. [5]European Spallation Source (ESS), Lund, Sweden. [6]Division of Computational Chemistry, Lund University, Lund, Sweden. ✉e-mail: stenmark@dbb.su.se

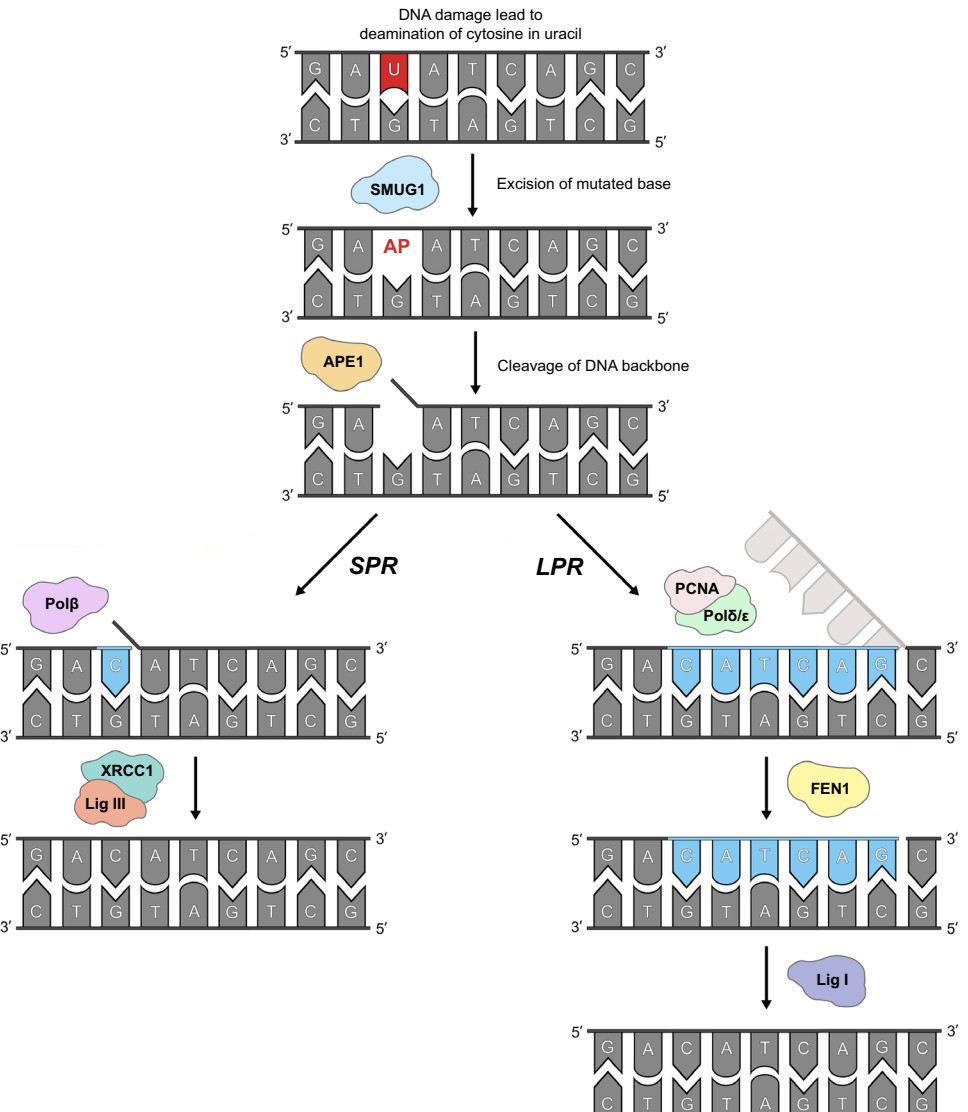

**Fig. 1 | Schematic overview of the hSMUG1-dependent BER pathway.** The repair process begins with the recognition and excision of mutagenic uracil (shown in red) from DNA by SMUG1, generating an abasic (AP) site. This AP site is processed further by APE1, which creates a nick in the phosphate backbone of the AP site; an essential step that allows enzymes involved in short-patch repair (SPR) or long-patch repair (LPR) to complete the BER process. Newly synthesised DNA bases added prior to ligation are shown in blue. Figure produced using Adobe Photoshop 2021 (version 22.5.1).

bifunctional glycosylases as the phosphate backbone is instead opened by β/δ-elimination[13]. In humans, a total of eleven DNA glycosylases have been characterised[14], of which three (hUNG, hSMUG1 and hTDG) belong to the uracil DNA glycosylase (UDG) superfamily[15]. Interestingly, all three of those enzymes are monofunctional glycosylases, however they differ in terms of their substrate specificities. For example, SMUG1 removes uracil, 5-hydroxymethyluracil (5hmU), 5-fluorouracil (5FU) and 5-hydroxyuracil (5-OHU) from both double-stranded DNA (dsDNA) and single-stranded DNA (ssDNA)[16,17], UNG exclusively removes uracil and 5FU from dsDNA and ssDNA[18], TDG excises uracil, 5hmU and T:G mismatches only from dsDNA[18]. 5FU is widely used in the treatment of numerous cancers, including colorectal, gastric, and breast cancer[19,20]. SMUG1 removes 5FU from DNA[17], and here we present the structural basis for this removal.

Gene knockout studies of *ung* and *smug1* showed an approximately 25-fold higher accumulation of 5hmU and uracil in DNA, highlighting the importance of these enzymes for genomic integrity[21]. However, it has been shown that while *msh2⁻/⁻ smug1⁻/⁻ ung⁻/⁻* knockout mice have a reduced lifespan, this is not the case for *msh2⁻/⁻ ung⁻/⁻* knockout mice, indicating that SMUG1 is able to compensate for a loss of UNG2 function[22]. hSMUG1 and hUNG share a low sequence identity of 15% and their cellular roles differ significantly. Specifically, there are two isoforms for hUNG (hUNG1 and hUNG2) which share a conserved catalytic domain and the binding motif for the nuclear ssDNA-binding protein RPA[23,24] but have different N-terminal sequences which localise hUNG1 and hUNG2 to the mitochondria and nucleus, respectively[25]. In contrast, there are no multiple isoforms of SMUG1 and the enzyme is only found in the cell cytosol and nucleus[26]. Furthermore, a 2002 study by Kavli et al. showed that hUNG2 is present at the replication fork whereas hSMUG1 is constitutively expressed and acts independently of the cell cycle[27].

In recent years, glycosylases have gained attention for their potential as diagnostic and therapeutic tools for cancer treatment[28,29]. Data from The Human Protein Atlas (https://www.proteinatlas.org) indicates a strong correlation between hUNG and hSMUG1 expression level and cancer survival probability. Furthermore, both low and high expression levels of SMUG1 and UNG have been correlated with either poorer or higher cancer survival probability, depending on the type of

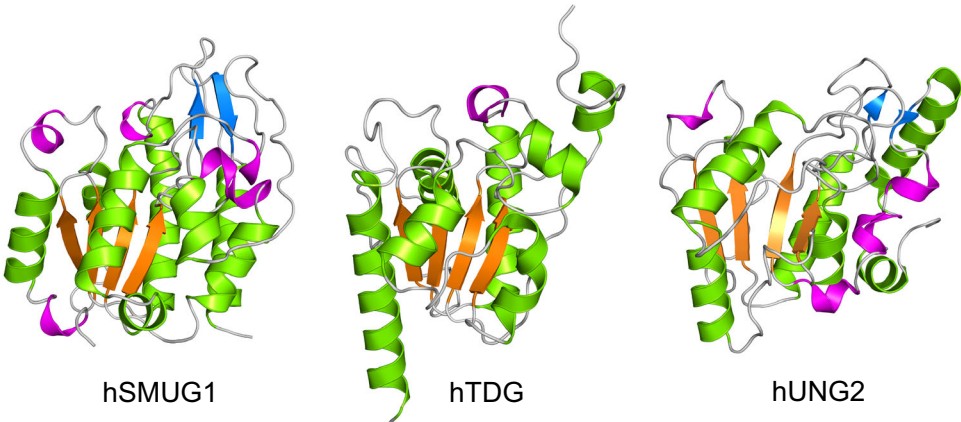

**Fig. 2 | Structure of hSMUG1 and other members of the UDG protein superfamily.** Cartoon representation of hSMUG1, Thymine DNA glycosylase (hTDG; PDB ID: 5FF8) and Uracil DNA glycosylase 2 (hUNG2; PDB ID: 5AYR) monomers, coloured according to secondary structure. Alpha-helices are coloured green, beta-strands are coloured blue, $3_{10}$-helices are coloured magenta and loop regions are coloured light grey. Figure produced with PyMOL (version 3.0.4, Schrödinger).

cancer, making both proteins interesting targets for drug development studies[30]. Rational structure-based drug design of enzyme inhibitors and/or activators relies on a solid understanding of an enzyme's structure and biochemical function. While several X-ray crystal structures of hUNG are available, no such information exists for hSMUG1.

Here, we present the structures of human SMUG1 including the apo form of the enzyme, complexes with the products uracil and 5FU, as well as the enzyme-product complex of hSMUG1 with dsDNA. Analysis of our product-bound structures, complemented with molecular dynamics (MD) simulations and neutron diffraction data, suggests the possibility of uracil rotation following base excision, whereas our dsDNA complex reveals how the uracil is flipped out and cut from the dsDNA. We identify three arginine residues that we verify to be key residues for both the DNA binding and the hSMUG1 enzyme activity. The structural information presented here will be vital for the future development of inhibitors and/or activators targeting hSMUG1.

## Results

### hSMUG1 is a member of the UDG superfamily of protein structures

We solved the structure of apo hSMUG1, determined to a high resolution of 1.38 Å. For all our structural studies, a truncated version of hSMUG1 lacking 24 amino acids at the N-terminus was used, as AlphaFold[31] predicted these residues to be an unstructured loop region (Supplementary Fig. 1). The hSMUG1 apo structure consists of eight α-helices, six β-strands and six $3_{10}$-helices (Supplementary Fig. 2). It adopts a core structural motif of a four-stranded parallel β-sheet flanked by α-helices, which is characteristic of members of the UDG protein superfamily such as hTDG and hUNG2 (Fig. 2)[32,33]. However, while these proteins share this conserved β-sheet motif, the surrounding structural elements differ significantly between the enzymes, which is reflected by the high RMSD values when the Cα-atoms of hSMUG1 are superimposed with hTDG (RMSD = 2.8 Å) and hUNG2 (RMSD = 4.3 Å) (Fig. 2).

### hSMUG1 binds to DNA via R124, R215 and region H239-K249

hSMUG1 was co-crystallised in the presence of a 12-base long uracil-containing dsDNA oligonucleotide identical to that used in a previously reported *Xenopus* SMUG1-dsDNA structure[16], selected to provide a stable duplex suitable for crystallisation. Visualisation of the hSMUG1 surface charge indicates that the dsDNA binds in a large positively charged area located on one side of the protein (Supplementary Fig. 3). Upon refinement, an AP site was observed at the position corresponding to uracil in the dsDNA sequence (Fig. 3A), consistent with the known high uracil excision activity of hSMUG1[34]

and rapid cleavage of the substrate during co-crystallisation. Closer inspection of the area surrounding the AP site indicates that hSMUG1 penetrates the double-helix via a DNA penetrating wedge, where residue R243 plays a major role (Fig. 3B). Comparison of hSMUG1-dsDNA with apo-hSMUG1 shows that, overall, the two structures superimpose very well, as indicated by a low RMSD value of 0.57 Å. There was however a significant difference for loop region H239-K249, close to the DNA AP site, which contains the DNA double helix penetrating residue R243 (Fig. 3C). Analysis of the hydrogen-bond (H-bond) network for the hSMUG1-DNA interaction indicates that the DNA backbone surrounding the AP site is supported by interactions with R124, S137, R187, R215, H239, S241 and N244 (Supplementary Fig. 4A). The AP sugar is positioned by a direct hydrogen bond with N176 and by a water-mediated hydrogen bond involving P86. In addition, the 3′ end of the AP site interacts with T178, while the 5′ phosphate forms hydrogen bonds with S137 and S241. In addition, the DNA strand opposite the AP site is positioned through interactions between the DNA bases and the helix-penetrating residues R243 and N248 (Supplementary Fig. 4B–D). When compared to apo SMUG1 it was evident that the main differences were for the side chains of residues R215, R243 and N248 (Fig. 3D).

In addition to the binding mode leading to catalysis observed in our hSMUG1-dsDNA structure, a non-productive DNA "end-binding" mode can also be observed when analysing the DNA interactions of monomers outside of the asymmetric unit (AU), which are related by crystallographic symmetry (Supplementary Fig. 5A). This non-productive partial interaction was previously observed in the structure of frog (*Xenopus laevis*) SMUG1 (xSMUG1) bound to DNA[16] (Supplementary Fig. 5B).

### SMUG1 product complexes suggest potential base rotation following cleavage

To gain insights into substrate binding, we soaked apo crystals of hSMUG1 with the products uracil and 5FU. Comparison of the hSMUG1-uracil and hSMUG1-5FU complexes with apo SMUG1 shows no significant differences in the overall structures, as indicated by the low RMSD values of 0.23 Å and 0.28 Å, respectively. Comparison of the products shows that they are located at the same position in the substrate binding site, where their respective pyrimidine rings overlap well (Fig. 4A). There was excellent, well defined electron density for both products in the active site (Fig. 4B, C). Uracil is positioned by H-bonds with the backbone nitrogen of M84 and the side chain atoms of N163 and H239. The pyrimidine base is additionally supported by a π-stacking interaction with F98 and an ordered water molecule, which is itself coordinated by the backbone nitrogen atoms of G87 and M91

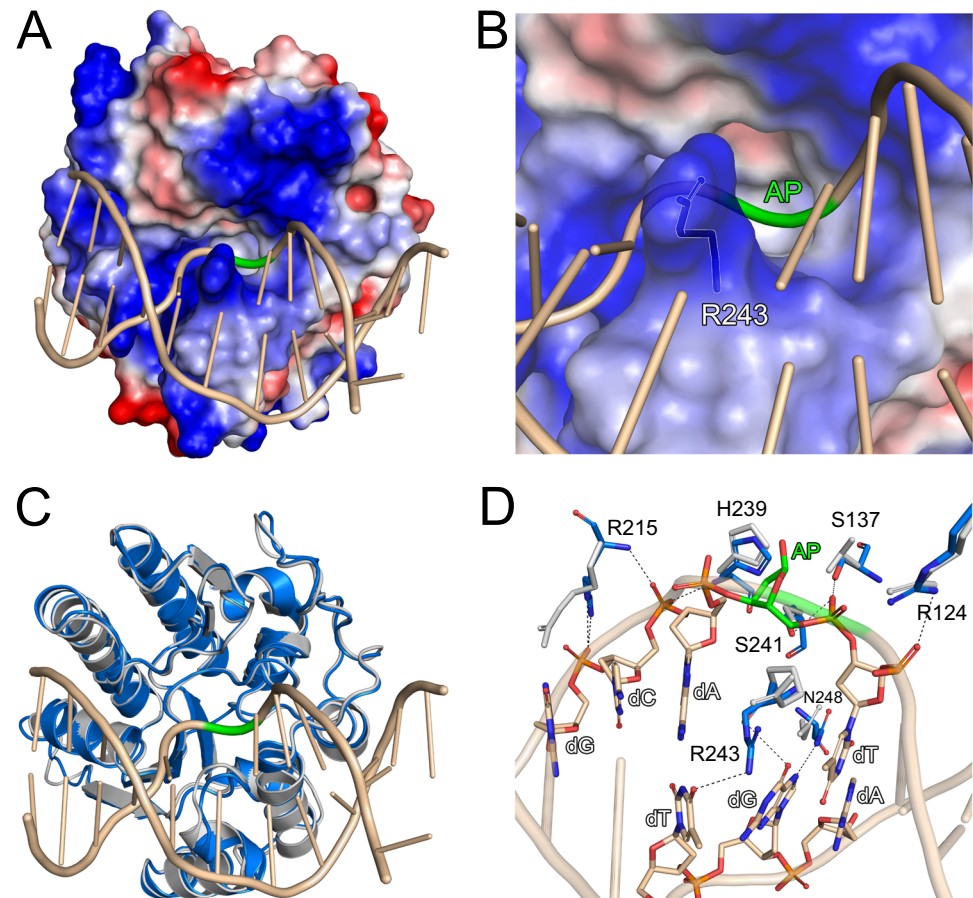

**Fig. 3 | Interaction of hSMUG1 with dsDNA. A** hSMUG1-dsDNA complex. The electrostatic potential of the hSMUG1 surface was calculated with APBS[97] in the range −5 kT (red, negative potential) to +5 kT (blue, positive potential). The dsDNA substrate is shown as a beige cartoon representation. The AP site in the dsDNA substrate is coloured green. **B** Zoomed-in view highlighting DNA-penetrating residue R243. The side chain of R243 is displayed as a stick model. **C** Comparison of hSMUG1-dsDNA (blue) with apo hSMUG1 (light grey) where protein monomers are displayed as cartoon representations. **D** H-bond network for DNA-interacting residues. Hydrogen bonds are shown as dashed lines. C atoms are coloured blue (hSMUG1-dsDNA), light grey (apo SMUG1), green (AP site of dsDNA) or beige (relevant deoxynucleotides), O atoms red, N atoms dark blue and P atoms orange. Deoxyguanosine (dG), deoxyadenosine (dA), deoxycytosine (dC) and deoxythymidine (dT) are shown as sticks. Figure produced with PyMOL (version 3.0.4, Schrödinger).

and the side chain oxygen of E135 (Fig. 4B). Superposition of the hSMUG1-uracil complex with xSMUG1 bound to both dsDNA and uracil (Supplementary Fig. 5C) shows strong conservation of the uracil binding residues, whereas the base itself is bound in opposite orientations, related by an approximately 180° rotation about the N3-C6 axis (Supplementary Fig. 5D). Analysis of the 5FU H-bond network shows that the fluorinated product is also positioned by the same interactions as uracil but also displays additional H-bonds between the fluorine of the base and the side chain atoms of S137 and H239 (Fig. 4C). Another point of difference is that two conformational states are observed for E135 in the uracil-bound structure; however, this is likely due to the significantly higher resolution of this structure (0.95 Å) compared to the 5FU complex (1.90 Å). Comparison with ligand-free hSMUG1 shows no differences in the active site binding residues, with the exception of the side chain of E135 which helps coordinate the aforementioned ordered water molecule (Fig. 4B, C). Superposition of the binary enzyme-product (hSMUG1-uracil) and hSMUG1-dsDNA complexes shows that the uracil is 3.4 Å away from the deoxyribose of the AP site (Fig. 4D), indicating the DNA-bound structure likely represents a partially relaxed state following uracil cleavage.

To generate a model of the hSMUG1-dsDNA complex prior to the excision of uracil, we performed MD simulations of this enzyme-substrate complex with the uracil flipped out of the dsDNA and positioned in the binding site. The complex was prepared using the crystal structure of the binary enzyme-product and three independent simulations of 300 ns each were performed. The trajectories were then clustered to identify interactions of uracil in the active site and complexes consistent with the catalytic mechanism. In the major cluster (56%), the flipped uracil base occupied a position similar to the free uracil product observed in the hSMUG1 crystal structure (Fig. 4E). In particular, the same set of five hydrogen bonds with residues M84, F98, N163 and H239 were formed, and we identified a water molecule at an appropriate distance to act as the catalytic nucleophile in more than half of the simulation snapshots. These results suggested that a catalytically competent state can be reached after minor rearrangements of the dsDNA and active site (Fig. 4E). While attached to DNA, the uracil can rotate around the *N*-glycosidic bond. However, when the DNA substrate is bound to hSMUG1, the uracil is locked in one orientation ("native" pose, Fig. 4E) as the rotation of the base is not feasible due to steric clashes in the active site. Following hydrolysis of the *N*-glycosidic bond, the free uracil can adopt an alternative orientation in the binding pocket due to the symmetry axis along N3 and C6 of the base. Despite our hSMUG1-uracil complex being determined at ultrahigh resolution (0.95 Å), it did not definitively indicate whether the uracil adopts the native or "rotated" orientation (Supplementary Fig. 6).

To further evaluate the two potential binding modes of uracil, we performed MD simulations combined with free energy calculations.

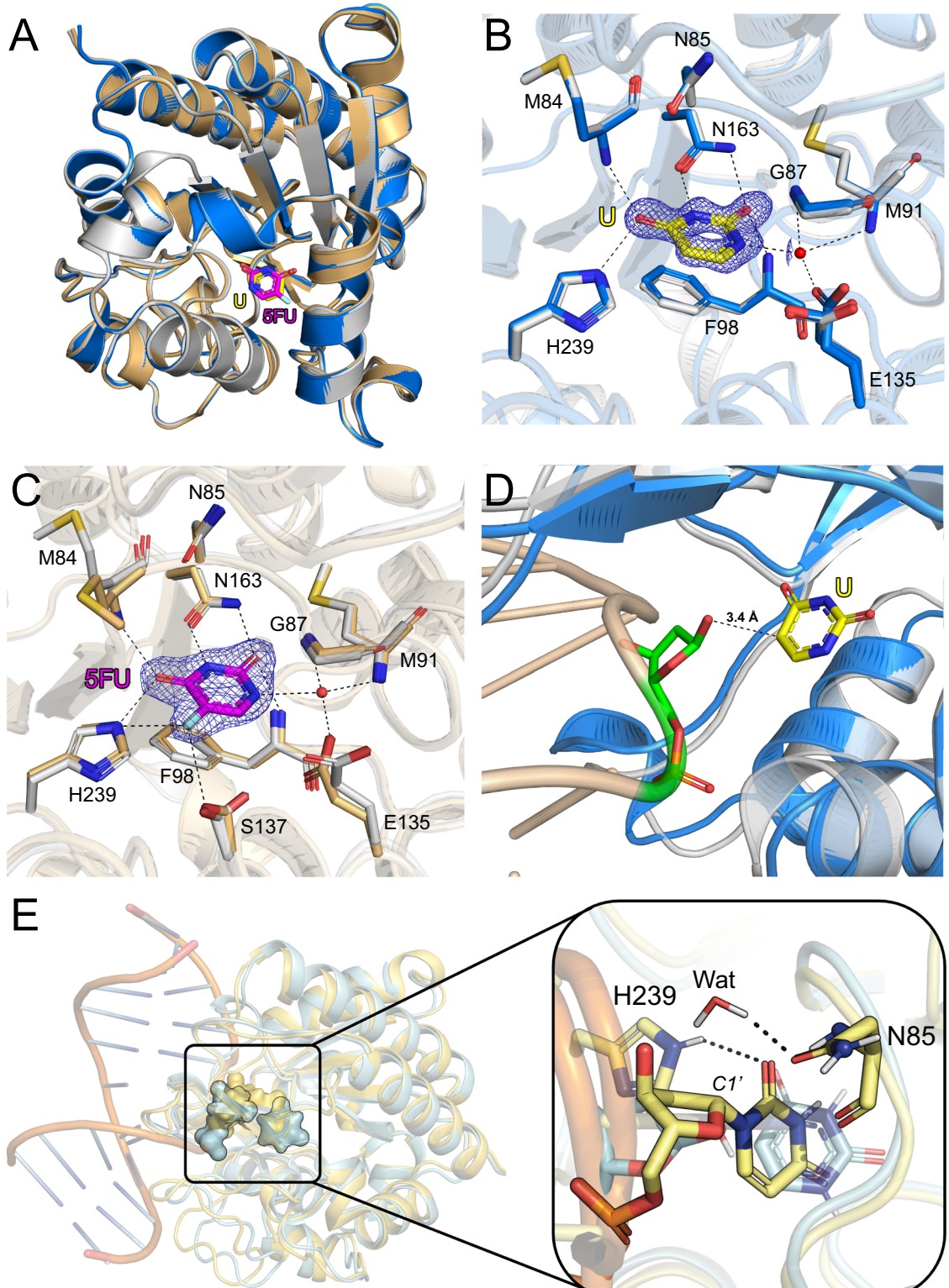

**Fig. 4 | hSMUG1 product complexes. A** Comparison of hSMUG1-apo (light grey) with hSMUG1-uracil (blue) and hSMUG1-5FU (light orange) where protein monomers are displayed as cartoon representations. Uracil (U) and 5FU are shown as sticks. C atoms are coloured yellow (U) or magenta (5FU), O atoms red, N atoms dark blue and F atoms cyan. **B** Uracil H-bond network. **C** 5FU H-bond network. In (**B** and **C**) the 2F$_o$-F$_c$ electron density maps around the ligand of interest are contoured at 1.8σ (uracil) and 1.3σ (5FU). H-bonds are depicted as black dashed lines. Water molecules are shown as red spheres. **D** Comparison of hSMUG1-dsDNA (light grey) with hSMUG1-Uracil (blue). The dsDNA substrate is shown as a beige cartoon representation. The AP site in the dsDNA substrate is coloured green. The distance between the AP site deoxyribose and the uracil product is indicated. **E** MD simulation snapshot of the hSMUG1-dsDNA complex with the uracil flipped out of the dsDNA into the active site. The most representative structure from the largest cluster (yellow) is aligned with the crystal structure of the hSMUG1-dsDNA complex (cyan), to which the free uracil (cyan) has also been added. hSMUG1-dsDNA is shown as a cartoon. The nucleotide, uracil, key residues, and a potential catalytic water from the MD simulations are shown as sticks. Hydrogen bonds are shown as dashed black lines. Figures produced with PyMOL (version 3.0.4, Schrödinger).

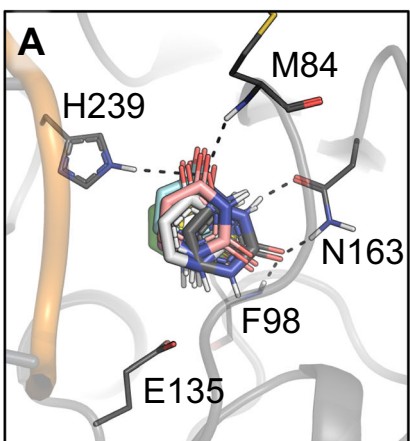
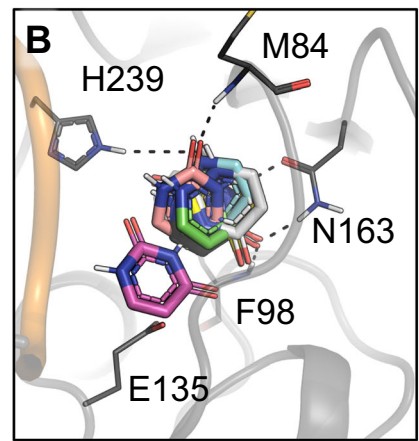
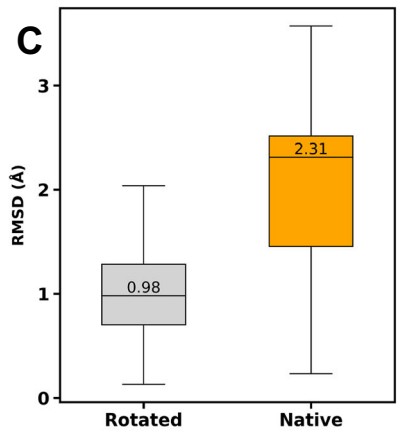

**Fig. 5 | Simulations of the ternary product complexes (hSMUG1-dsDNA-uracil) with the free uracil in rotated and native poses.** Representative structures from six independent simulations (green, cyan, magenta, yellow, pink and white) of **A** the rotated and **B** the native poses are aligned based on the protein structures to the initial conformation (reference, dark grey). Key binding site residues are labelled and shown as lines. Key interactions between uracil and M84, F98 and N163 from the starting binding poses are indicated with black dashed lines. **C** RMSD values for uracil from the last 100 ns for rotated (grey) and native (orange) poses. Median values for both systems are shown in each box. The box represents the upper and lower quartiles, with whiskers extending to 1.5x the interquartile range.

The relative binding free energy between the rotated and native poses in the binary enzyme-product complexes was computed using free energy perturbation (FEP) (Supplementary Fig. 7). These calculations showed that the rotated binding pose of uracil was energetically more favourable than the native pose by 1.59 ± 0.16 kcal/mol (Supplementary Table 1). We also complemented these calculations with MD simulations of the hSMUG1-dsDNA complex with the uracil product bound in the native and rotated poses (ternary product complexes). Based on six simulations of 300 ns for each complex, the rotated pose was more stable than the native orientation, with median root-mean-square deviations (RMSD) values of 0.98 and 2.31 Å, respectively. The rotated pose maintained the hydrogen bonding with H239, N163, M84, and F98 observed in the crystal structure in all simulations (Fig. 5A), whereas the interactions with N163 and M84 were lost in four simulation replicas of the native pose (Fig. 5B). The rotated orientation of uracil is also supported by our hSMUG1-5FU structure. As 5FU does not have a symmetry axis along N3 and C6 due to its additional fluorine atom, this clearly indicates 5FU can adopt a rotated orientation following base excision (Supplementary Fig. 8).

To further clarify the protonation states of key catalytic residues and the orientation of the uracil in the active site of hSMUG1, we performed neutron diffraction experiments on the H/D-exchanged crystals of hSMUG1 in complex with perdeuterated uracil (Supplementary Fig. 9A, B). The density is weak for the C5 carbon of uracil in the neutron scattering density maps (Supplementary Fig. 9C). However, the density is strong in most other regions of the active site. The density supports the rotated orientation, with a strong hydrogen bond between a water molecule (wat1) and the N1 atom of the uracil. The density clearly shows that the side chain Nε2 of H239 is protonated and hydrogen bonded to the O4 in uracil. In addition, the main chain nitrogen atoms of F98 and M84 are hydrogen bonded to the O2 and the O4 atom of uracil, respectively. The orientation and protonation of N163 can be clearly determined with its two pronounced hydrogen bonds to the uracil. The orientation and protonation state of N85, which alongside H239 is a key catalytic residue[35], is also observed in the density map. The hydrogen bonding network around wat1 observed in the neutron scattering density map led to several possible donor-acceptor solutions. MD simulations were performed to understand the hydrogen bonding network in this region. In these calculations, restraints on the backbone heavy atoms were applied in order to keep the structure close to that observed in the neutron diffraction experiments. The unrestrained uracil maintained the key interactions

with M84, F98, N163 and H239 in the simulations. The simulations supported that wat1 accepts a hydrogen bond from the N1 atom of uracil and the backbone amide of M91. Moreover, one hydrogen of wat1 was interacting with the side chain carboxylate of E135, while the second hydrogen did not form any hydrogen bonds (Supplementary Fig. 9D). Overall, the use of neutron crystallography enabled direct determination of the protonation states of key catalytic residues within the active site, together with the protonation state of the uracil base. These observations provide structural support for a model in which uracil undergoes a rotation following cleavage from the DNA backbone.

## SMUG1 preference for uracil over thymine
MD simulations combined with free energy calculations were used to investigate the selectivity of hSMUG1 for uracil over thymine, one of the four canonical DNA bases. We first performed simulations for the binary enzyme-substrate complex containing thymine instead of uracil. Whereas the interactions with key residues M84, F98, N163 and H239 were maintained in the simulations with the uracil substrate, introduction of the thymine led to displacement of the base and none of the five hydrogen bonds were formed. For example, uracil formed stable hydrogen bonds with N163, with an occupancy of 80%, whereas the same interaction was observed in less than 10% of the simulations with thymine (Supplementary Fig. 10). In a second step, the relative binding free energy of the uracil substrate relative to thymine in the native pose was calculated following the same protocol used for two uracil binding poses (Supplementary Table 2). The calculated relative binding free energy was 1.54 ± 0.15 kcal mol⁻¹, corresponding to a preference for uracil over thymine. Comparison of simulation snapshots of the complexes with uracil and thymine showed that the methyl group of thymine clashes with the side chain of E135 and displaces the ordered binding site water molecule identified in the crystal structure.

## hAPE1 outcompetes hSMUG1 from dsDNA following uracil cleavage
Early activity studies of hSMUG1 showed that the enzyme is subject to product inhibition on dsDNA following uracil cleavage[36]. In order for the BER process to progress, hAPE1 needs to release hSMUG1 from DNA, after which it cleaves the DNA backbone adjacent to the AP site (Supplementary Fig. 11A). To gain insights into how hAPE1 releases hSMUG1 from dsDNA following product inhibition, we co-crystallised hSMUG1 together with hAPE1 in the presence of a uracil-containing

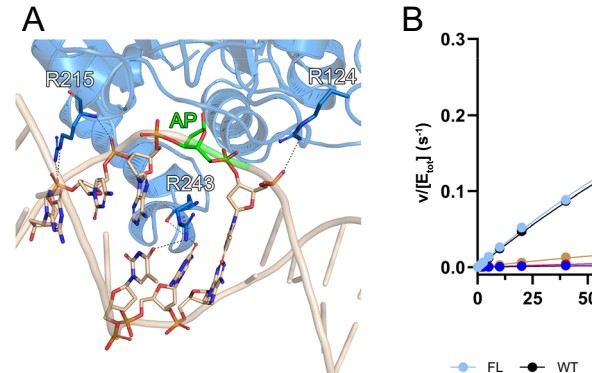

**Fig. 6 | Single-point hSMUG1 arginine-to-alanine mutants show greatly reduced activity. A** hSMUG1-dsDNA structure (blue) highlighting DNA interacting arginine residues R124, R215 and R243. dsDNA is shown as a beige cartoon. Relevant deoxynucleotides are shown as sticks. Hydrogen bonds are shown as black dashed lines. C atoms are coloured blue (hSMUG1-dsDNA), green (AP site of dsDNA) or beige (deoxynucleotides), O atoms red, N atoms dark blue and P atoms orange.

Figure produced with PyMOL (version 3.0.4, Schrödinger). **B** Fluorescence-based hSMUG1 activity assay saturation curves. FL: Full-length hSMUG1. WT: N-terminally truncated hSMUG1-Δ24, used as "wild-type" protein for enzyme assays, crystallography and MST studies. Experiments were performed twice with data points in triplicate. Data are presented as mean ± SD ($n = 2$).

oligonucleotide, which has a non-hydrolysable backbone due to the presence of phosphorothioate modifications (Supplementary Tables 3 & 4). This resulted in the structure of hAPE1 in complex with dsDNA containing an AP site (Supplementary Fig. 11B). Analysis of the electron density clearly shows the backbone next to the AP site is completely intact, a result of the aforementioned phosphorothioate modifications (Supplementary Fig. 11C). Here, the absence of hSMUG1 in the crystal structure indicates that, under the high concentrations used for crystallisation, hAPE1 can occupy AP-sites in dsDNA with higher apparent affinity than hSMUG1. While this structure represents a product-bound intermediate, it does not provide direct insight into the mechanism by which hAPE1 and hSMUG1 coordinate or compete, and multiple turnover events in solution may also contribute to enzyme dynamics. It is unlikely that hAPE1 could cleave the DNA backbone while SMUG1 is bound, given that AP nucleotide flipping into the APE1 active site is required for catalysis. When comparing our AP-site containing APE1-dsDNA complex with previously reported APE1-dsDNA structures, it is unique in incorporating an abasic sugar bearing a C1′ hydroxyl group rather than the tetrahydrofuran (THF) analogue commonly used to mimic AP sites[37–39]. This feature provides a closer chemical approximation to a native AP; however, the additional 3′- and 5′-end phosphorothioate modifications flanking the AP-site mean that the substrate is not fully physiological. Overall, the protein-DNA architecture and active-site organisation remain consistent with earlier structures (Supplementary Fig. 11D, E), indicating that inclusion of the C1′ hydroxyl does not alter the established mode of APE1-dsDNA recognition.

## hSMUG1-arginine-to-alanine mutants lose affinity and activity towards dsDNA

The hSMUG1-dsDNA complex highlighted several DNA-interacting residues, of which three arginines (R124, R215, R243) were deemed to be particularly important due to their close proximity to the AP site (Fig. 6A). In order to determine the effect these residues have on hSMUG1 enzyme activity and on DNA binding affinity, we produced hSMUG1-R124A, hSMUG1-R215A and hSMUG1-R243A single-point mutants and analysed them using a fluorescence-based activity assay coupled with APE1 (Supplementary Fig. 12) and microscale thermophoresis (MST). Fluorescence signal was converted to concentration formed product using a standard curve produced by full conversion of the dU oligo duplex substrate at different concentrations to product by adding an excess of enzymes and by following the reaction over time (Supplementary Fig. 12A). As the hSMUG1Δ24 "wild-type"

construct we used for crystallographic studies lacks an unstructured loop at its N-terminus, we also tested the activity of full-length hSMUG1 (Supplementary Table 5, Supplementary Fig. 13A). Importantly, the activity of the slightly truncated hSMUG1 construct was similar to that of the full-length enzyme, indicating that the aforementioned N-terminal region does not influence hSMUG1 catalysis. Notably, all of the single-point arginine mutants were significantly less active compared to wild-type hSMUG1 (Fig. 6B, Supplementary Fig. 13B, Supplementary Table 5). Since substrate saturation could not be obtained for the SMUG R124A or R215A mutants, neither $k_{cat}$ nor $K_m$-values could be determined. However, apparent second order rate constants ($k_{cat}/K_m$) were determined from the slope of the Michaelis-Menten curve at low substrate concentrations enabling comparison of their catalytic efficiencies. R124A displays a 60-fold lower $k_{cat}/K_m$-value compared to the wildtype while the corresponding figure for R215A is 25-fold lower. The R243A mutant displays a 7-fold lower $k_{cat}/K_m$ value compared to the wild-type enzyme. For this mutant the lower $k_{cat}/K_m$ value seems to be mostly due to a considerably lower $k_{cat}$-value (6-fold) rather than a higher $K_m$-value.

To determine whether the reduced activity of the arginine mutants could be due to their effect on DNA binding, we used MST to assess their binding affinities with various dsDNA and ssDNA substrates (Table 1, Supplementary Fig. 14). This included unmodified substrates, as well as oligonucleotides containing uracil or an AP site. It should be noted that as wildtype hSMUG1 is catalytically active, MST measurements with cleavable substrates yield apparent affinities that may reflect a mixture of enzyme bound to the intact substrate and enzyme bound to the excised product. Wild-type hSMUG1 was shown to have the highest affinity for dsDNA containing uracil and 5FU but had poorer affinity for dsDNA containing an AP site. The low-nanomolar apparent affinity observed from uracil-containing dsDNA likely represents binding under catalytic conditions, whereas the weaker affinity measured for AP-site dsDNA may partly result from the use of a tetrahydrofuran abasic-site analogue, which is chemically distinct from a natural AP-site (Supplementary Fig. 15). Consistent with this interpretation, the catalytically inactive mutant hSMUG1-N85A[35,40] bound uracil containing dsDNA with an affinity indistinguishable from wildtype (Supplementary Table 6), demonstrating that tight binding does not require uracil excision during the measurement. The affinity was significantly weaker (three orders of magnitude lower) for uracil-containing ssDNA and for both unmodified dsDNA and ssDNA substrates. Compared to wild-type hSMUG1, the hSMUG1-R124A, hSMUG1-R215A and hSMUG1-R243A mutants all showed significantly weaker

**Table 1 | Dissociation constants (Kd) for hSMUG1 variants binding to single- and double-stranded DNA substrates measured by MST**

| hSMUG1 variant | | | | | |
|---|---|---|---|---|---|
| # | Oligonucleotide | WT Kd (nM) | R243A Kd (nM) | R215A Kd (nM) | R124A Kd (nM) |
| 1 | No modification, dsDNA (5'-CGGACTCACGGG-3') | 1267 ± 49 | 2339 ± 129 | NB | 2435 ± 231 |
| 2 | dU, dsDNA (5'-CGGACTUACGGG-3') | 7 ± 0.5 | 290 ± 18 | 512 ± 28 | 2455 ± 364 |
| 3 | 5FU, dsDNA (5'-CGGACT/i5F-dU/ACGGG-3') | 8 ± 0.6 | 540 ± 78 | 605 ± 71 | 2852 ± 230 |
| 4 | AP site, dsDNA (5'-CGGACT/idSp/ACGGG-3') | 66 ± 4 | 604 ± 33 | 5512 ± 724 | 2008 ± 148 |
| 5 | No modification, ssDNA (5'-CCCGTGAGTCCG -3'Cy5) | 1305 ± 56 | 3293 ± 263 | NB | 3030 ± 92 |
| 6 | dU, ssDNA (5'-CGGACTUACGGG -3'Cy5) | 1081 ± 122 | 3021 ± 175 | NB | 2593 ± 111 |

AP-site DNA refers to a tetrahydrofuran (THF) abasic-site analogue (dSpacer, /idSp/).
NB, no detectable binding under the conditions tested. WT denotes the N-terminally truncated hSMUG1-Δ24 construct, which was used throughout this study as the wild-type protein for crystallography, enzymatic assays, and MST measurements. Kd and standard deviations (mean values ± SD) were calculated using fits from at least three individual titrations.

binding affinities with all tested substrates (Table 1). As observed for the wild-type enzyme, the mutants had the highest affinity for dsDNA-uracil and dsDNA-AP substrates, and displayed significantly lower affinity for ssDNA-uracil, the unmodified dsDNA and ssDNA oligonucleotides. Both wild-type and hSMUG1-N85A bound uracil-containing ssDNA weakly (Supplementary Table 6), indicating that ssDNA is a poor binding substrate and that substrate turnover does not dominate the observed ssDNA binding behaviour. Interestingly, hSMUG1-R215A showed no detectable binding to unmodified dsDNA or ssDNA substrates. hSMUG1-R215A displayed consistently low affinity for all substrates tested, whereas hSMUG1-R243A showed higher affinity towards uracil and AP site containing dsDNA compared to the other mutants consistent with the higher catalytic efficiency observed for this mutant compared to the other mutants.

## Discussion

Herein we present the structures of human SMUG1, including the SMUG1-DNA complex captured in a catalytically competent state. The dsDNA substrate binds within a broad, positively charged groove, while the opposite face of the protein exhibits distinct negatively charged regions. Given that SMUG1 has been shown to be involved in telomere processing and to directly interact with Dyskerin pseudouridine synthase 1 (DKC1) as part of RNA processing[41,42], these negatively charged surfaces may mediate electrostatic interactions with protein partners such as DKC1. The oligonucleotide in our hSMUG1-dsDNA structure contains an AP site, consistent with the rapid and well characterised cleavage of the *N*-glycosidic bond[34] and release of uracil by hSMUG1 during co-crystallisation. Comparison with our uracil bound structure shows the distance between the base and the AP site is too far to form a covalent *N*-glycosidic bond. Our DNA complex thus likely represents a relaxed, product inhibited, state following uracil cleavage. Consistent with prior predictions[16], our DNA-bound structure reveals a DNA-penetrating loop in hSMUG1 and defines its boundaries to residues H239-K249.

Studies of DNA-binding proteins have shown that arginine residues are heavily enriched in areas required for phosphate and base interactions with the DNA backbone[43]. To cleave the *N*-glycosidic bond, hSMUG1 has to first flip out the damaged base from the DNA double helix into the active site, which our DNA-bound structure suggests, in line with findings by Wibley et al.,[16] is performed by arginine residue R243. Such a base-flipping mechanism has previously been studied for the glycosylase UNG2[44–47]. We also identified two additional arginine residues, R124 and R215, which we proposed were

particularly important for DNA binding due to their close proximity to the AP site. The single-point arginine-to-alanine mutants of these residues were all significantly less active and had much weaker affinity for DNA substrates compared to wild-type hSMUG1. Interestingly, this result disagrees with a 2007 study where a hSMUG1-R243A mutant was shown to be nearly as active as the wild-type enzyme[40]. Once uracil is flipped into the binding pocket, hydrolysis of the *N*-glycosidic bond is proposed to occur via a dissociative $S_N1$-like mechanism[35]. Previous mutagenesis studies have revealed N85 and H239 are the catalytic residues which are critical for the cleavage of the *N*-glycosidic bond[35,40]. Our hSMUG1 product complexes with either uracil or 5FU in context of our dsDNA bound structure, indicates that N85 and H239 are located within the uracil binding pocket, whereas R124, R215 and R243 are located outside this deeper pocket, on the surface of the protein. The greatly diminished activity of these arginine mutants is therefore likely not due to any effects on the actual catalysed hydrolysis itself, but rather the consequence of secondary effects. In addition to their role in ensuring the correct positioning of DNA prior to base cleavage, it is possible that these arginine residues may be involved in the scanning mechanism of hSMUG1 when it is sliding along dsDNA to identify uracil bases.

Our MST data showed that the apparent affinity of each single-point mutant for DNA substrates was significantly weaker than for wild-type hSMUG1. The kinetic analysis showed that the $K_m$ value of the R243A mutant is similar to that of the wildtype enzyme. Since $K_m$ is a composite of reaction rate constants for the formation and breakdown of the enzyme substrate complex ($K_d$) and $k_{cat}$, this shows that not only substrate binding is affected by the R243A mutation (as shown by MST) but also other steps on the way to product release. This is reflected in a $k_{cat}$-value that is approximately 6-fold lower and a $k_{cat}/K_m$-value that is 7-fold lower compared to the wildtype enzyme. Since we could not obtain any substrate saturation for the R124A and the R215A mutants and consequently could not determine $k_{cat}$ and $K_m$ values, we determined $k_{cat}/K_m$ values from the initial slope of the saturation curves. The R124A and the R215A mutants display catalytic efficiencies ($k_{cat}/K_m$) that is 60-fold and 25-fold lower compared to the wildtype enzyme, respectively. For these mutants this is consistent with the poor substrate binding, observed by MST, resulting in reduced enzymatic activity. The $k_{cat}$-value determined for hSMUG1 wt here (0.8 min⁻¹ or 0.95 min⁻¹ dependent on hSMUG1 construct) is very close to what has previously been reported (0.84 min⁻¹)[35]. A similar value ($k_{obs}$=0.4 min⁻¹ (0.0067 s⁻¹)) was reported in another study[48]. As noted earlier, the poor enzymatic activity observed for the hSMUG1-

R243A mutant differs significantly to a previous report, in which this variant displayed similar activity to wildtype SMUG1[40]. This discrepancy may be due to the fact that the activity of hSMUG1 R243A in that publication was measured at a single substrate concentration of 20 nM using a radioactively labelled substrate, whereas in the present study, a fluorophore and quencher labelled substrate was used and the activity was monitored at concentrations ranging from 0 to 160 nM. The addition of a fluorophore and quencher on the substrate is likely to reduce substrate affinity and hence increase $K_m$. The enzyme is probably saturated at 20 nM of the radioactively labelled substrate based on the reported $K_m$ of 2 nM for wild-type SMUG1[35]. Thus, differences in activity between the R243A mutant and wildtype SMUG1 at lower non-saturating substrate concentrations would not be observed under these conditions.

Early activity assay studies of hSMUG1 showed that the enzyme is subject to product inhibition on dsDNA after base cleavage[36]. This product inhibition is relieved by hAPE1, which displaces SMUG1 from the DNA, which greatly accelerates SMUG1 activity[40]. Based on this knowledge, it can be assumed that our hSMUG1-dsDNA complex is likely the product inhibited state. In this structure, there is no uracil in the substrate binding pocket, indicating that it has diffused away after substrate cleavage. This may be due to the low concentration of uracil in the DNA substrate, relative to the concentration of free uracil that was soaked into hSMUG1 apo crystals to obtain the hSMUG1-uracil complex.

Notably, when hAPE1 was included with hSMUG1 during co-crystallisation, only the structure of hAPE1 bound to AP-containing dsDNA was obtained. This likely reflects the higher apparent affinity of hAPE1 for AP-sites under the crystallisation conditions, rather than revealing the mechanism by which hAPE1 accesses the DNA. Consistent with this, previous studies have shown that hAPE1 can relieve hSMUG1 product inhibition even when its endonuclease activity is blocked[36].

Interestingly, our hSMUG1-5FU product-bound structure showed that 5FU adopted a conformation which suggests that the free base is capable of rotating within the active site pocket following hydrolysis of the *N*-glycosidic bond. This finding was supported by our free energy calculations, which indicated that this alternative orientation is more energetically favourable than the native conformation of uracil when it first enters the deeper substrate binding pocket. Due to the product inhibition of hSMUG1 on dsDNA following uracil cleavage, rotation of the uracil base could be part of the proteins mechanism to prevent the catalysis of the backward reaction, thereby preventing the uracil being reattached to the DNA (Supplementary Fig. 16). Additionally, our MD simulations supported that the native binding pose is less stable than the alternative one, further supporting the notion that uracil rotation helps to suppress the backward reaction. However, while our simulations define the thermodynamic preference for this orientation, they do not address the kinetics of the rotation and therefore cannot distinguish whether this occurs before or after dissociation of the abasic DNA.

Thymine is identical to uracil with the exception that it contains a methyl group at the 5-position of the pyrimidine ring (Supplementary Fig. 17A). Our crystal structures of hSMUG1 bound to uracil and to 5FU offer insights into the subtle structural and chemical features that could underlie this exquisite substrate discrimination. Comparison of our rotated uracil and 5FU-bound complexes showed subtle differences in binding. Both 5FU and the water molecule, which binds the ligand is shifted slightly relative to the uracil complex (Supplementary Fig. 17B). This likely results from the fluorine atom at the 5-position of the pyrimidine ring (Supplementary Fig. 17A). Our MD simulations of the binary enzyme-substrate complex provided mechanistic insight into hSMUG1 selectivity by revealing how subtle chemical differences between uracil and thymine translate into distinct interaction patterns and binding stability. Superimposing thymine with uracil in both the rotated and native binding poses suggests that thymine binding may

be disfavoured in both orientations in the binary enzyme-product complexes. If thymine were to adopt the rotated conformation, its methyl group would come into close proximity to H239, likely resulting in a steric clash not observed with 5FU (Supplementary Fig. 17C). Furthermore, when bound in the native orientation, our MD simulations show that the 5-methyl group of thymine displaces the aforementioned water molecule in the binding pocket (Supplementary Fig. 17D). This water is also present in the apo structure, implying that it plays a structurally important role. However, the fluorine of 5FU would also clash with this water when the ligand binds in the native orientation, suggesting it is possible for this water to be displaced by the electronegative fluorine and then adopt this original position following base flipping. Notably, the fluorine atom in 5FU is not only smaller (having a much smaller van der Waals radius) but is also more electronegative, which may allow it to be accommodated without disrupting the local environment. In contrast, the hydrophobic methyl group on thymine may be energetically unfavourable in both conformational contexts. Taken together, these findings suggest a model in which hSMUG1 discriminates against thymine through a combination of steric hindrance and unfavourable chemical interactions. Such selectivity is essential, given that thymine is a canonical DNA base, and its inappropriate removal would have harmful consequences for genome integrity.

As noted earlier, members of the UDG superfamily differ in their tolerance for C5 substitutions of uracil. Superposition of the binding pocket in our SMUG1–uracil structure with the UNG2–dsDNA-uracil complex (PDB ID: 1EHM[49]) shows that hUNG2 and SMUG1 share key features involved in uracil recognition, including a histidine and an asparagine positioned similarly in the active site (Supplementary Fig. 18). However, in the binding mode demonstrated by Parikh et al., larger C5 substituents would be sterically restricted by a tyrosine residue Y147, suggesting a more constrained pocket in UNG2. Although UNG2 has been reported to recognise 5FU, the structural basis for this tolerance remains unclear[40]. Comparison of SMUG1–uracil with a hTDG-dsDNA-5hmU complex (PDB ID: 4FNC[50]) indicates a markedly different mode of substrate accommodation. The TDG active site contains a largely hydrophobic pocket formed by L124, A145 and P153, whereas the corresponding region in SMUG1 is more polar and positively charged (Supplementary Fig. 18). This difference in chemical environment may contribute to the distinct range of tolerated C5 modifications. While the hydrophobic TDG pocket could disfavour substituents such as 5-OHU or 5FU, its ability to process 5hmU suggests that additional factors beyond simple hydrophobicity influence substrate selection.

In conclusion, we present the structures of hSMUG1, including an enzyme-product complex with dsDNA as well as with the reaction products uracil and 5FU. In the context of our enzyme activity data, MST analysis and FEP calculations, this provides key insights into the DNA repair mechanism for this medically important glycosylase. The structural insights presented here provide a valuable foundation for future efforts to develop targeted inhibitors or activators of hSMUG1 and other DNA glycosylases.

## Methods

### Experimental methodology

**Annealing of dsDNA oligos.** All oligonucleotides used in this study were ordered from GenScript, and were purified via HPLC by the manufacturer. Some oligonucleotides used for MST analysis were additionally labelled with Cy5 at the 3′ end (Supplementary Table 3). To prepare double-stranded oligos (dsDNA) the lyophilised powders were first dissolved in buffer containing 50 mM HEPES pH 7.5, 150 mM NaCl, 5 % glycerol to a final concentration of 1 mM. The relevant sense and antisense strand for each dsDNA oligo were mixed together in a 1:1 ratio and then annealed by incubation at 95 °C for 5 min after which the temperature was gradually lowered by 1 °C every 40 s, until the

temperature reached 4 °C. For crystallography experiments, dsDNA-uracil and dsDNA-NHAP oligonucleotides were made and for MST analysis dsDNA-Cy5, dsDNA-uracil-Cy5 and dsDNA-AP-Cy5 oligos were prepared (Supplementary Table 4).

**Plasmids.** Codon optimised *H. sapiens* APE1 DNA encoding residues 39-318 for wild-type APE1 (hAPE1) and *SMUG1* DNA encoding residues 25-270 for wild-type SMUG1 (hSMUG1) and the mutants N85A (hSMUG1-N85A), R124A (hSMUG1-R124A), R215A (hSMUG1-R215A), H239A (hSMUG1-H239A) and R243A (hSMUG1-R243A) were each cloned into a pET-24a(+) vector between *NdeI* and *XhoI* restriction sites (GenScript). The resulting expressed proteins have an N-terminal His-tag and TEV protease cleavage site followed by the protein of interest. Full-length human SMUG1 was expressed from pE-SUMOpro-SMUG1 (a kind gift from Dr Mark Thompson, Sheffield University) as SMUG1 N-terminally fused to 6xHis-SUMO. The ULP-1 protein was used to remove the SUMO tag and was expressed from pHYRS52[51] (a gift from Hideo Iwai (Addgene plasmid # 31122)).

**Protein expression and purification.** All proteins in this study were expressed and purified in an identical manner using the following procedure: His-tagged hSMUG1, hSMUG1-N85A, hSMUG1-R124A, hSMUG1-R215A, hSMUG1-H239A, hSMUG1-R243A and hAPE1 were expressed in *E. coli* BL21(DE3) cells for 20 h at 18 °C after induction with 0.5 mM IPTG. Cultures were grown in TB media supplemented with 50 µg/mL kanamycin, using a LEX Bioreactor system. Cells were harvested and resuspended in lysis buffer (100 mM HEPES pH 7.5, 500 mM NaCl, 10 mM imidazole, 10% glycerol and 0.5 mM TCEP), following which the cells were lysed via sonication. The lysates were clarified by ultracentrifugation at 42,000 $g$ for 1 h and the resulting supernatant was syringe-filtered using a 0.45 µm PES filter (Sarstedt). The clarified lysate was loaded on to an ÄKTA pure system (Cytiva) and purified via immobilised metal affinity chromatography using a 5 mL HisTrap HP column (Cytiva). Fractions containing the protein of interest were pooled together and dialysed in dialysis buffer (20 mM HEPES pH 7.5, 200 mM NaCl) for 6 h after which TEV protease was added to the dialysis tube using a protein:protease molar ratio of 1:50. The sample was then left to dialyse overnight at 4 °C. Following His-tag cleavage, the sample was loaded onto a 5 mL HisTrap HP column and the flow-through fractions containing tag-free protein were collected. The relevant protein was further purified by gel filtration using a HiLoad 16/600 Superdex 200 pg column (Cytiva) equilibrated with gel filtration buffer (20 mM HEPES, 300 mM NaCl, 10 % glycerol, 0.5 mM TCEP). The purified protein samples were verified to be >95 % pure by SDS-PAGE. Aliquots of pure hSMUG1 proteins and hAPE1 were flash frozen in liquid nitrogen, and stored at -80 °C.

ULP-1 and Hisx6-SUMO-SMUG1, generating full-length SMUG-1, were expressed in Rosetta (DE3) pLysS (Novagen #70956-4). Expression was induced by addition of 0.5 mM IPTG at $OD_{600} = 0.7$. Bacteria were incubated with shaking at 25 °C overnight followed by lysis of the bacteria using ultrasonication in 100 mM HEPES pH 8.0, 500 mM NaCl, 10% glycerol, 10 mM imidazole, Benzonase (25 U/ml) and 1x Cocktail Protease Inhibitors (Roche). Lysates were cleared by centrifugation and filtered through a 0.45 µm filter before being loaded onto a 5 ml His-Trap HP column (Cytiva) equilibrated with Buffer A (20 mM HEPES, 500 mM NaCl, 25 mM imidazole and 10% Glycerol, 2 mM DTT pH 8.0). Protein was eluted by increasing the imidazole concentration from 25 mM to 500 mM. For ULP-1 purification, fractions containing ULP-1 were pooled and dialysed against 50 mM Tris-HCl pH 7.5, 150 mM NaCl, 10% Glycerol and 0.5 mM TCEP and further purified using a Superdex 75 16/600 column (GE Healthcare). Fractions from the His-Trap HP column containing the His-SUMO-SMUG1 protein were pooled and dialysed against 150 mM Tris-HCl pH 8.0, 2 mM DTT. 100 µg ULP-1 was added to the His-SUMO-SMUG1 fusion protein and incubated for 2 h at room temperature to cleave off the SUMO-tag. Thereafter, SMUG-1 was

purified using a Superdex 75 16/600 column (Cytiva) equilibrated with gel filtration buffer (20 mM HEPES, 300 mM NaCl, 10 % glycerol, 0.5 mM TCEP). The purified protein samples were analysed for purity using SDS-PAGE and the purified proteins were aliquoted, flash frozen and stored at -80 °C.

**Crystallisation.** To obtain ligand-free hSMUG1 crystals (hSMUG1-apo), purified hSMUG1 (26.5 mg/mL) was mixed in a 1:1 ratio with 0.1 M Buffer System 3 pH 8.4, 80 % (v/v) P500MME_P20K, 0.12 M Alcohols Mix (Morpheus Screen, Molecular Dimensions) and crystallised via sitting-drop vapour diffusion at 21 °C. Large rod-shaped crystals appeared within 1 day and were fished without additional cryoprotectant and flash-cooled in liquid nitrogen. SMUG1-uracil complex crystals were obtained by soaking hSMUG1-apo crystals in crystallisation growth solution supplemented with 15 mM uracil, for 4 h prior to fishing and cryo-cooling. For the hSMUG1-dsDNA complex, hSMUG1 (23.7 mg/ml) was mixed in a 1:1 ratio with 800 µM of dsDNA-uracil, a uracil-containing dsDNA oligo. The mixture was incubated at room temperature for 2 h and then concentrated so that the final hSMUG1 concentration was 23.7 mg/ml. The resulting protein-DNA solution was then mixed in a 1:2 ratio with 0.1 M diammonium hydrogen citrate pH 4.9, 60% w/v PEG 3350 and then crystallised via sitting-drop vapour diffusion at 21 °C. Rod-shaped crystals appeared within 1 week. For the hAPE1-DNA complex, hAPE1 (26.2 mg/mL) and hSMUG1 (23.7 mg/ml) were mixed in a 1:1:1 ratio with 950 µM of dsDNA-uracil*, a dsDNA oligo where the uracil base is flanked on each side by a phosphorothioate modification. The mixture was incubated at room temperature for 2 h. The resulting protein-DNA solution was then mixed in a 1:1 ratio with 0.15 M ammonium sulphate, 0.1 M Na-HEPES pH 7.0, 20% w/v PEG 4000 and then crystallised via sitting-drop vapour diffusion at 21 °C. Diamond-shaped crystals appeared within 1 day.

**Structure determination.** X-ray diffraction data were collected on the BioMAX beamline at MAXIV (Lund, Sweden) and stations I03, I04 and I24 of the Diamond Light Source (Oxford, UK). The datasets were collected at 100 K at a wavelength of either 0.95373 Å (hSMUG1-apo), 0.6491 Å (hSMUG1-uracil), 0.9795 Å (hSMUG1-5FU), 0.6702 Å (hSMUG1-DNA) or 0.9093 Å (hAPE1-DNA). Single crystals were used for data collection, with the exception of the hSMUG1-uracil structure where data from two crystals were collected and later merged. Data for hSMUG1-apo were indexed and integrated using DIALS[52], followed by scaling using AIMLESS[53] within the CCP4 suite[54]. The structure was solved via molecular replacement (MR) with Phaser[55] using the monomer of *Xenopus laevis* SMUG1 (PDB ID: 1OE4) as the search model. Several rounds of model building and refinement were performed using Coot[56] and REFMAC[57] during which waters and ligands were incorporated into the structure. For the hSMUG1 complexes, data were processed using DIALS[52] (hSMUG1-uracil) or STARANISO[58] (hSMUG1-DNA). The structures were solved via MR with Phaser using our hSMUG1-apo structure with ligands and waters removed, as the search model. Model building and refinement were then carried out using Coot[56] and either phenix.refine[59] (hSMUG1-uracil) or REFMAC[57] (hSMUG1-DNA and hSMUG1-5FU). Data for the hAPE1-DNA structure were indexed and integrated using DIALS[52], followed by scaling using AIMLESS[53]. The structure was solved via MR with Phaser[55] using the monomer of *H. sapiens* APE1 (PDB ID: 4QHD) as the search model. Model building and refinement were performed using Coot[56] and REFMAC[57]. Data collection and refinement statistics are presented in Supplementary Table 7. The coordinates and structure factors for hSMUG1-apo, hSMUG1-uracil, hSMUG1-5FU, hSMUG1-DNA and hAPE1-DNA were deposited in the PDB under the accession codes 9GGS, 9GK0, 9RQP, 9GM2, and 9RQS respectively.

**Microscale thermophoresis.** The binding of hSMUG1 and the variants hSMUG1-N85A, hSMUG1-R124A, hSMUG1-R215A and hSMUG1-R243A

to DNA was studied using microscale thermophoresis (MST)[60]. All DNA oligos were prepared as described in the Annealing of dsDNA oligos Methods section. MST was performed using the Monolith NT.115 instrument (NanoTemper Technologies GmbH). Binding was measured between the SMUG1 proteins and the Cy5 labelled single and/or double-stranded oligonucleotides (Supplementary Tables 3 & 4) using MST buffer containing 20 mM HEPES pH 7.5, 10% glycerol, 50 mM KCl, 2 mM DTT and 0.05% Tween-20. A 16-step serial dilution of each protein was prepared by adding 10 µl MST buffer to 15 PCR tubes. A 20 µl volume of SMUG1 protein (9.2-9.5 µM) was added to the first tube, and 10 µl was then transferred to the second tube and mixed thoroughly by pipetting (1:1 dilution series). To each tube of the dilution series, 10 µl of the relevant single or double-stranded Cy5 labelled DNA oligonucleotide was added to reach a final concentration of 5 nM. The samples were incubated for 5 min at room temperature before being transferred to Monolith™ NT.115 Series Capillaries (NanoTemper Technologies, GmbH). The capillaries were scanned in the MST instrument at 40 % excitation power, medium MST power. Cy5 labelled single-stranded (ssDNA-Cy5) and double-stranded (dsDNA-Cy5) oligos without any mutation were included as negative controls. The binding of each oligo to the various SMUG1 proteins was performed in triplicate.

**Fluorescence-based hSMUG1 enzyme activity assay.** The activity of the N-terminally truncated hSMUG1Δ24 was assayed in a coupled fluorescence-based assay and compared with the activity of the full-length protein. The substrate for the hSMUG1 assay was prepared by annealing a 5′ end FAM-labelled uracil-containing oligo (5′-FAM-TCTGCCAUCACTGCGTCGACCTG), with a 1.25-fold excess of a 3′ end Dabcyl-labelled oligonucleotide (5′-CAGGTCGACGCAGTGGTGG-CAGT-Dabcyl) resulting in a DNA duplex with a U:G mismatch. The oligonucleotides were purchased from ATDBio. The activity of 1 nM SMUG1 was tested in assay buffer (50 mM HEPES pH 7.4, 100 mM KCl, 7.5 mM MgCl$_2$, 5% glycerol, 0.0022% Tween 20, fish gelatin (1:1000, Sigma/Aldrich), 0.4 mM DTT) containing 2 nM APE1, added in excess to cleave the formed AP site. Activity was assayed at substrate concentrations ranging from 0 to 160 nM by monitoring the increase in FAM fluorescence (Exc485/Em535) over time (Supplementary Fig. 19) using a Hidex Sense Multimode plate reader. To convert fluorescence to concentration formed product a standard curve was produced by fully converting various concentrations of the U containing substrate, ranging from 0 to 60 nM, to product by adding an excess of enzymes (10 nM SMUG1 and 10 nM APE1). Maximal fluorescence was plotted against substrate concentration (Supplementary Fig. 12B) and linear regression was performed using GraphPad Prism 8.0. Initial rates were calculated, after subtraction of fluorescence of wells containing APE1 but not hSMUG1, using linear regression and kinetic parameters were determined by fitting the Michaelis-Menten equation to the initial rate data points using GraphPad Prism 8.0. Experiments were performed twice with data points in triplicate.

**Neutron diffraction experiments.** Purified hSMUG1 (26.5 mg/mL) was mixed in a 1:1 ratio with 0.1 M Buffer System 3 pH 8.4, 80 % (v/v) P500MME_P20K, 0.12 M Alcohols Mix (Morpheus Screen, Molecular Dimensions) and crystallised via sitting-drop vapour diffusion at 21 °C. A seed stock was then prepared from a single crystal containing 0.1 M Buffer System 3 pH 8.4, 50 % (v/v) P500MME_P20K, 0.12 M Alcohols Mix (Morpheus Screen, Molecular Dimensions). 50 µL of hSMUG1 (21.5 mg/ml) was mixed with 5 µL of 1:100000 diluted hSMUG1 seedstock and 45 µL of crystallisation solution (0.1 M Buffer System 3 pH 8.4, 50 % (v/v) P500MME_P20K, 0.12 M Alcohols Mix, 30 % D$_2$O). The 100 µL drop was equilibrated over 20 mL crystallisation solution via sitting-drop vapour diffusion at 21 °C. After three weeks the crystals stopped growing, reaching a final size of 2x1x0.5 mm. The crystal was then soaked for 4 h with 15 mM uracil-d$_2$ (MedChemExpress) dissolved in deuterated DMSO, P20K dissolved in D$_2$O and 0.1 M

tris and 0.1 M bicine pD 8.8 dissolved in D$_2$O. The crystal was then mounted in a quartz capillary for data collection.

Neutron quasi-Laue diffraction data for the hSMUG1-uracil complex were collected at room temperature (19 °C) using the LADI-III diffractometer[61] at the Institut Laue-Langevin (ILL) in Grenoble, France. Data were obtained from a crystal with an approximate volume of 1 mm³. A neutron wavelength band of 2.8–3.8 Å (Δλ/λ = 30 %) was used, yielding diffraction data to a resolution of 2.3 Å. During data collection, the crystal remained stationary at different φ angles (around a vertical rotation axis) for each exposure. A total of 30 diffraction images were recorded (12 h per exposure) from two different crystal orientations. The images were indexed and integrated using the LAUEGEN[62] software. The wavelength normalisation was performed with LSCALE[63], using the intensities of symmetry-equivalent reflections measured at different wavelengths. The data were then merged and scaled using SCALA[53].

Room-temperature X-ray data were collected from the same crystal used for neutron data collection at a wavelength of 0.9795 Å. The X-ray data were recorded on the BM07-FIP2 beamline at the European Synchrotron Radiation Facility (ESRF) in Grenoble, France. The data were processed using the XDS software[64], scaled and merged with AIMLESS[65], and converted to structure factors using TRUNCATE from the CCP4 suite[54]. The structure was solved by molecular replacement with Phaser[55] using our apo hSMUG1 model. Modelling was carried out using Coot[56] and the model was refined using the X/N joint refinement option of phenix.refine[59]. Data collection and refinement statistics are presented in Supplementary Table 8 and the structure was deposited to the PDB under the accession code 9SQ2.

**Computational chemistry**
**Molecular dynamics simulations.** Three systems were prepared for MD simulations: (1) hSMUG1-dsDNA with the uracil flipped out of the dsDNA into the active site (enzyme-substrate complex), representing the pre-cleavage state of the glycosidic bond, (2) hSMUG1-dsDNA with the free uracil product in its native binding pose, and (3) hSMUG1-dsDNA with the free uracil product in a rotated binding pose. All systems were derived from the crystal structure of hSMUG1-dsDNA containing an AP site. The free uracil was added to the hSMUG1-dsDNA complex by superimposing the structure of hSMUG1 in complex with 5FU (hSMUG1-5FU). The fluorine atom was removed, and uracil was positioned either in the rotated or native binding pose to form the ternary product complexes. For the enzyme-substrate complex, a thymidine monophosphate was aligned to the sugar moiety from the crystal structure using PyMOL (version 3.0.4, Schrödinger), and the methyl group was deleted to yield uracil monophosphate. The simulation of hSMUG1 in the absence of DNA and with the free uracil bound in the rotated conformation was generated using the neutron diffraction structure.

**System preparation and simulation details: MD simulations.** Uracil force field parameters were generated using Antechamber from AmberTools23[66]. Atom types and partial charges were assigned using the GAFF2 force field and the AM1-BCC method, respectively[67,68]. The required bonded parameters were generated using parmchk2 from AmberTools23[66]. For the free uracil, atom types for the deoxyribose-phosphate moiety at the AP site were assigned to match those of a canonical deoxyribose-phosphate as defined in the OL15 force field from Amber23[69,70]. Partial charges were determined using the Restrained Electrostatic Potential (RESP) method at the HF/6-31 G* level[71,72] using ORCA v6.0.1[73,74]. An equivalent protocol was applied to parametrise the deoxyribose-phosphate moiety bound to uracil in the enzyme-substrate complex. Topology and coordinate files for all systems were prepared with the LEaP module from AmberTools23[66]. The ff14SB[75] and OL15[69] force fields were used for protein and DNA, respectively. Systems were solvated with the TIP3P water model[76]

using a 15 Å padding distance. Water molecules within 1 Å of the solute were removed to avoid steric clashes. The crystallographic waters buried in the protein interior and involved in hydrogen bonding with uracil were retained. A physiological salt concentration of 0.15 M NaCl was added to the system.

All simulations of the binary enzyme-substrate and ternary hSMUG1-dsDNA-uracil complexes were carried out using AMBER23[70]. Each system was first energy minimised with 10,000 steps of conjugate gradient under restraints of 10.0 kcal/mol·Å² on all non-solvent atoms, followed by 10,000 steps with a reduced restraint force constant of 5.0 kcal/mol·Å². Subsequently, 10,000 steps were performed with backbone restraints of 5.0 kcal/mol·Å², and an additional 10,000 steps with the restraint reduced to 1.0 kcal/mol·Å². Finally, an unrestrained minimisation of 10,000 steps was performed. The systems were then heated from 0 to 300 K in 30 K increments, every 20 ps, using the Langevin thermostat with a collision frequency of 2.0 ps⁻¹. The equilibration consisted of a 1 ns simulation under the NVT ensemble, followed by a 5 ns simulation under the NPT ensemble with isotropic position scaling to maintain 1 atm via the Berendsen barostat[77] with a relaxation time of 2 ps. During equilibration, positional restraints on protein, DNA, and ligand heavy atoms were progressively reduced from 10 to 1 kcal/mol·Å². Finally, NPT equilibrations for 1 ns with restraints only on the backbone atoms, followed by 1 ns of unrestrained equilibration, were performed. Long-range electrostatic interactions were treated with the particle mesh Ewald (PME) method[78]. The resulting structures were used as starting points for multiple replicas of 300 ns under the NPT ensemble, which were initialised with different random seeds. Trajectory frames were stored every 10 ps. A nonbonded cutoff of 10 Å was used, and all bonds involving hydrogen atoms were constrained using the SHAKE algorithm[79] with a tolerance of 10⁻⁶ Å and integration time step of 2 fs. Identification of the most representative conformations for each system was performed by aligning the simulation snapshots to the protein Cα atoms, followed by clustering based on the uracil heavy atoms using the average linkage algorithm implemented in the *cpptraj* module of AmberTools23[66,80,81]. RMSD was calculated relative to the first frame, after aligning to the Cα atoms of the protein (excluding terminal residues), using the *cpptraj* module of AmberTools23[66,82].

**Free energy calculations.** Relative binding free energy (ΔΔG) between the rotated (A) and native (B) predicted binding poses computed using an alchemical thermodynamic cycle protocol[83] (Supplementary Fig. S7):

$$\Delta\Delta G = \Delta G^{bound}_{A-B} - \Delta G^{unbound}_{A-B} \qquad (1)$$

where $\Delta G^{bound}_{A-B}$ and $\Delta G^{unbound}_{A-B}$ denote the free energy differences between binding poses A and B in the hSMUG1–uracil complex (bound state) and in bulk water (unbound state), respectively. Since the end states in the unbound system are chemically identical, $\Delta G^{unbound}_{A-B}$ is expected to be zero. Similarly, the binding free energy difference between uracil (U) and thymine (T) was computed as:

$$\Delta\Delta G = \Delta G^{bound}_{U-T} - \Delta G^{unbound}_{U-T} \qquad (2)$$

where $\Delta G^{bound}_{U-T}$ and $\Delta G^{unbound}_{U-T}$ correspond to the alchemical transformation from uracil to thymine in the bound and unbound states, respectively. All alchemical free energy simulations were performed using the FEP method in NAMD version 3.0b6[84] using the OPLS-AA/M force field[85]. Detailed descriptions of the simulation setup, λ-windows, soft-core parameters, and convergence criteria are provided below.

**System preparation and simulation details: FEP calculations.** The initial binary enzyme-product complex (hSMUG1-uracil) was generated from the hSMUG1-5FU crystal structure by removing the fluorine

atom. Protein preparation was carried out using Schrödinger's Protein Preparation Wizard[86,87]. In this step, protonation states of titratable residues at pH 7 were assigned using PROPKA[88], hydrogen atoms were added, and water molecules orientations within 5 Å of uracil were refined. Uracil parameters were generated using LigParGen[89], providing OPLS-AA force field parameters and 1.14*CM1A-LBCC partial charges[68,90]. Topology and coordinate files for both bound and unbound states were generated using *psfgen* in VMD[91]. Both systems were solvated using TIP3P water molecules[76] with a 15 Å padding in each direction and an NaCl concentration of 150 mM was added. Crystallographic water molecules involved in hydrogen bonding with uracil and buried waters in the protein interior were retained.

Simulations were carried out using NAMD 3.0b6[84]. Systems were initially minimised using 1000 steps of conjugate gradient to resolve steric clashes. Following minimisation, the systems were gradually heated from 0 to 300 K in 10 K increments over a total of 80 ps using a Langevin thermostat[92], with a damping constant of 1 ps⁻¹. Equilibration was carried out in two stages: 1 ns under the NVT ensemble, followed by 0.5 ns under the NPT ensemble via the Nosé-Hoover barostat to maintain 1 atm, with a piston oscillation period of 100 fs and a damping constant of 50 fs[93,94]. Backbone atoms were harmonically restrained in these steps with a force constant of 10 kcal/mol·Å², with restraints progressively relaxed over a total of 5 ns. This was followed by a final 0.5 ns equilibration phase with all restraints removed. A nonbonded cutoff of 12 Å was applied, with switching and pair distances of 10 Å and 14 Å, respectively. Long-range electrostatics were calculated with particle mesh Ewald with a grid spacing of 1.0 Å[78]. All bonds involving hydrogen atoms were constrained using the SHAKE algorithm[79] with a tolerance of 10⁻⁶ Å and integration time step of 2 fs.

Following equilibration, FEP calculations were performed using the dual-topology approach. In the unbound systems, each transformation (A to B) used 20 equally spaced λ-windows to interpolate between the two states, from λ = 0 (state A) to λ = 1 (state B), with a step size of 0.05 per window. For the bound systems, 70 equally spaced λ-windows were used. Each window started with 5000 steps of energy minimisation, followed by heating of the system, and four steps of 0.2 ns of equilibration, in which harmonic restraints from the backbone heavy atoms were progressively relaxed. In the following step, 0.5 ns of unrestrained equilibration and 5 ns of production under the NPT ensemble were performed. Particles from state B were gradually introduced (appearing) while those from state A were simultaneously removed (annihilated). Electrostatic interactions for appearing particles were fully turned off at λ ≤ 0.5, and linearly turned on from λ > 0.5 to λ = 1. Conversely, electrostatics for disappearing particles were fully on at λ = 0 and linearly turned off, becoming completely decoupled by λ ≥ 0.5. Van der Waals (vdW) interactions for appearing particles were off at λ = 0, and linearly turned on to λ = 1. For annihilated particles, vdW interactions started fully on at λ = 0 and decreased linearly, becoming fully off at λ = 1. A soft-core vdW radius-shifting coefficient of 6.0 was used to prevent particle overlapping during the transformation. Five independent replicas were generated for the bound state and three for the unbound state. Additionally, reverse transformations (from state B to state A) were carried out using the same protocol. Equivalent heavy atoms between states A and B were manually defined, and a harmonic bias with a force constant of 5.0 kcal/mol·Å² was applied to maintain spatial proximity between corresponding atoms using the Collective Variables (Colvars) module in NAMD[95]. These restraints were symmetrically applied in both the bound and unbound states to avoid contributions to free energy differences.

## Reporting summary
Further information on research design is available in the Nature Portfolio Reporting Summary linked to this article.

## Data availability

The crystallographic and neutron diffraction data generated in this study has been deposited to the RCSB Protein Data Bank under the accession codes 9GGS, 9GK0, 9GM2, 9RQS, 9RQP and 9SQ2. Raw data for the affinity and activity studies are provided within the Source Data File. The input files required to perform the MD simulations, together with initial and final coordinates, have been deposited at Zenodo (https://doi.org/10.5281/zenodo.19453249)[96]. Source data are provided with this paper.

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

## Acknowledgements

We thank Rohit Kumar, Stockholm University, for his valuable advice on the preparation of samples for neutron diffraction experiments. We thank Ronja Gottschewski for assistance with protein purification and Henrietta Nielsen, Stockholm University, for allowing us to use the MST instrument. We acknowledge the ESRF (Grenoble, France) for provision of synchrotron radiation beamtime on beamline BM07-FIP2, supported by the French ANR PIA3 (France 2030) EquipEx+ project MAGNIFIX under grant agreement ANR-21-ESRE-0011. The authors would like to thank Diamond Light Source for beamtime (proposal mx29948), and the staff of beamlines I03, I04 and I24 for assistance with crystal testing and data collection. We acknowledge the MAX IV Laboratory for beamtime on the BioMAX beamline under proposal 20220278. Research conducted at MAX IV, a Swedish national user facility, is supported by Vetenskapsrådet (Swedish Research Council, VR) under contract 2018-07152, Vinnova (Swedish Governmental Agency for Innovation Systems) under contract 2018-04969 and Formas under contract 2019-02496. This work was supported by grants from the Swedish Cancer Society (24-3694 to T.H., 22 2473 Pj and 25 4860 Pj to J.C. and 24 3848 Pj to P.S.), the Swedish Research Council (2015-00162 to T.H. and 2022-03681 to P.S.), the Swedish Childhood Cancer Foundation (2024-0048) to T.H., EU-IMI2 (875510), Wallenberg Scholars (2023.0225), the Cancer Research Funds of Radiumhemmet (221123), the Marie Skłodowska-Curie PRISMAS programme (funding ID: 101081419) to P.S and the Swedish Strategic Research Programme eSSENCE to J.C. The computations were enabled by resources provided by the National Academic Infrastructure for Supercomputing in Sweden (NAISS), partially funded by the Swedish Research Council through grant agreement no. 2022-06725.

## Author contributions

J.M.L. and P.S. designed the study. J.M.L., E.S.H., A.S.J., E.W., and E.W. expressed and purified wildtype and mutant proteins. S.S. performed and analysed the microscale thermophoresis experiments. A.S.J. and E.W. produced and analysed the fluorescence based activity assay data under supervision of O.M. and T.H. J.M.L. produced the protein crystals for X-ray and neutron experiments, collected and processed X-ray data and produced the X-ray and neutron structures under supervision of E.S.H. and P.S. S.A. supervised the neutron crystallography experiment. L.G. collected and processed the neutron diffraction data. G.V.J. performed the MD simulations and analysed the results together with S.P. under the supervision of J.C. and I.C. J.M.L., E.S.H., G.V.J., P.S., and A.S.J. wrote the manuscript with input from the other authors.

## Funding

## Competing interests

The authors declare no competing interests.
