## [Transparent Peer Review file · Nature Communications]

Structural basis for uracil removal from DNA by human SMUG1

Corresponding Author: Professor Pål Stenmark

Version 0:

Reviewer comments:

Reviewer #1

(Remarks to the Author)

This manuscript from Stenmark and coworkers reports the first crystal structures of human SMUG1, an important DNA glycosylase that protects against mutations by excising damaged bases including uracil, 5-hydroxyuracil, and other lesions. High resolution structures are presented for the free enzyme, the enzyme bound to uracil and 5-F-uracil, and the enzyme bound to the AP DNA product (without the cleaved U base). Prior structures of SMUG1 were solved for *Xenopus laevis* SMUG1 bound to DNA. The structures and MD simulations of SMUG1 bound to U and 5FU indicate that the base binds in an unexpected orientation, when compared to the prior SMUG1 structures or those of the related enzyme UNG2. suggested the possibility of an interesting and unanticipated mechanism whereby the excised base flips within the active site. The manuscript also reports glycosylase activity and DNA-binding affinity for wild-type SMUG1 and several mutants, though there are issues with the approach and interpretation of these studies. Overall, the results could be of substantial interest to those the field of DNA damage and repair, but the following concerns should be addressed prior to potential publication.

lines 43-46: the statements “complex networks of DNA repair, also known as DNA damage response (DDR)”, and “the base excision repair (BER) pathway... is the predominant DDR pathway for repairing small base lesions” should be revised. The DDR and DNA repair pathways including BER are distinct processes; BER is not typically considered as a “DDR pathway”

line 58: APE1 has endonuclease activity, which is distinct from AP lyase activity

line 65: MBD4 is not a member of the UDG superfamily. It belongs to the Endo IV superfamily that also includes MutY. Also, on line 72 “MBD” should be MBD4

lines 104-5: “the structure of SMUG1 in a productive complex with dsDNA”. For an enzyme (such as SMUG1), a “productive complex” typically refers to an enzyme-substrate complex that is poised for the chemical step to occur (which is halted by an enzyme mutation or use of a non-reactive substrate analog). The structure reported here should be referred to as an enzyme-product complex.

Lines 151-3: “Upon refinement, it was evident that there was an AP site at the position where uracil was located in the dsDNA sequence, indicating the hSMUG1 successfully cleaved the uracil from the DNA substrate during co-crystallization.” The phrasing sounds as though this result might not have been expected. SMUG1 has very high activity for U excision (kobs ~200 min⁻¹; Stivers group, 2011, *Biochemistry* 50:618-27) so the AP site is generated as soon as the enzyme and DNA have been mixed, particularly at the concentrations used for crystallization. This should be reworded accordingly, in this section and other sections noting this observation.

Fig. 3. The R243 side chain is hard to see in panel B. I may be informative to show more structural detail in panel B, or add another panel with more detail (e.g., stick for AP site and relevant SMUG1 groups, water molecules). It would be helpful to include a simple cartoon figure that shows all of the enzyme-DNA duplex interactions, including phosphate contacts in an easily viewed manner. Fig 3D indicates no contacts with the AP sugar or the phosphate on its 3' side. Is this correct and are there any water-mediated contacts? If so, they should be shown and if not, this should be discussed.

Fig 4. It would be informative to include a figure that shows the alternative (or both) orientations of U in the active site and shows the enzyme interactions, particularly because the orientations cannot be distinguished by electron density.

It would be informative to show H bonds in panel E, perhaps for a representative snapshot from the MD simulations.

Lines 225-8: The manuscript states that, "While attached to DNA, the uracil can rotate around the N-glycosidic bond. However, the simulation showed that when the base enters the hSMUG1 active site pocket, the uracil is locked in one orientation ("native" pose). Following hydrolysis of the N-glycosidic bond, the free uracil can adopt a "flipped" orientation in the binding pocket due to the symmetry axis along N3 and C6 of the base". This needs some clarification. Does the simulation indicate the U base is "locked" in the orientation shown in 4E for the modeled enzyme-substrate complex (prior to bond cleavage)? Was a simulation of the ternary product complex performed that shows the U can adopt the orientation in panel 4B without dissociation of the abasic DNA? Or is this conclusion based solely on the structure or simulations of the binary (enzyme-U) complex? This is an important mechanistic question that needs clarification and potentially additional MD studies. It would be helpful to use the terms "enzyme-substrate complex" or "lesion recognition complex" when referring to structures/models of SMUG1 bound to DNA with an intact N-glycosyl bond (rather than "uracil covalently linked"), as these terms are typical in the literature for structures of DNA glycosylases.

Given that nucleotide flipping is mechanism used by DNA glycosylases to extrude the substrate nucleotide out of the DNA duplex, and many readers will associate the terms "flipped" or "flips" with this process. For example, in the abstract, the statement that "the substrate uracil flips following base excision" could confuse many readers. As such, it may be better to use a different term for the potential alternative conformation of the uracil base suggested by the new structures, in this section and throughout the manuscript.

For the MD simulations described on p. 12 of the ternary product complex (SMUG1-DNA-U), was the U found to change its pose during the MD simulation? While the results indicate that the ternary complex with the alternate U pose is more stable, it is important to clarify whether the results provide evidence that the change from the "native" pose (expected just after bond cleavage) and the alternative pose can occur in the ternary product complex, or if it requires dissociation (and rebinding) of the AP DNA.

Lines 298-307. The finding that SMUG1 binds U tighter than T, based on MD simulations is interesting, but additional simulations modeling DNA-bound SMUG1 with an intact dT substrate flipped into the active site would be much more informative and could be compared to that shown in Fig 4E for dU.

Lines 321-3. Regarding the structure of DNA-bound APE1 resulting from a crystallization sample that contained a SMUG1 substrate and both SMUG1 and APE1, this is an end state that could be reached by many potential mechanisms. I think the only conclusion to be drawn is that under the conditions used for crystallization (very high concentrations of both enzymes and DNA), that APE1 has higher affinity for its substrate (AP DNA) than does SMUG1 for its product. I think the statement that, "the absence of hSMUG1 shows that hAPE1 releases hSMUG1 from the AP site containing dsDNA by outcompeting hSMUG1 prior to backbone cleavage by hAPE1, rather than by cleaving the DNA backbone first in order to facilitate hSMUG1 release", should be revised to avoid the suggestion of an active mechanism by APE1. It could be active, but this can't be ascertained by the structure. Also, given that APE1 acts by flipping the AP nucleotide into its active site, it seems impossible that it might be able to do so when SMUG1 is bound to the AP site. I think the latter part of this sentence should be revised because it isn't really a plausible mechanism given what is known about APE1.

Regarding the APE1-AP-DNA structure, which mimics the enzyme-substrate complex, it would be informative to briefly discuss how this structure advances, if at all, upon prior structural and mechanistic results for APE1.

Fig 6B. The activity data for w/t and mutant SMUG1 is shown and apparently fitted to the MM equation but the resulting parameters (k_{cat} , K_m , and k_{cat}/K_m) are not provided. These need to be included in a table and discussed.

Fig 6B (cont.). The duration of the experiments appears to be too short by ~twofold for wild-type enzyme and much too short for the mutants to give a reliable fit to the MM eqn; this is a hyperbolic eqn and the experiments need to be collected at a high enough substrate concentration to saturate the enzyme. This is not the case for the wild-type enzyme and cannot be readily judged for the mutants because the y-axis scale is too large. The mutant results should be shown in a different panel with a y-axis that is small enough in range to see whether the substrate concentration is high enough to give a reliable fitting for V_{max} .

Fig 6B (cont.) The main text should state that the enzymatic assay is a coupled assay, where the fluorescence signal is produced by a second enzyme, APE1, which acts on the product (AP DNA) generated by SMUG1. Also, controls are needed to demonstrate that the reactions are collected under conditions such that the observed activity is dependent entirely on the activity of SMUG1 rather than APE1. The activity of SMUG1 is relatively high and the observed change in fluorescence could be due in part to APE1.

Fig 6B (cont.) Ideally, the units for the y-axis should be converted to product concentration per min rather than fluorescence change per min. This would allow the results to be compared quantitatively to previous studies of human SMUG1, which should be included in the manuscript.

Fig 6B (cont.). It would be informative to show the raw data from the fluorescence assay for at least one enzyme-DNA combination, including for all substrate concentrations used for fitting the results in Fig 6B, in the main text or Supplementary Information.

The DNA binding experiments and results need some clarification. Given the very high activity of SMUG1 for U and 5FU, the binding assay for wild-type enzyme is likely measuring SMUG1 affinity to its product(s), depending on the concentration of

enzyme and DNA substrate. Regarding the latter, the Methods section indicates a 5 mM final concentration of DNA was used, which seems much too high. Perhaps it was 5 nM? Controls are needed to assess the extent to which DNA substrates are converted to AP DNA product during the timeframe of the experiment, for w/t and mutant enzymes. These details should be clarified at the beginning of this topic in the results section (line ~350) and in Table 1, and throughout the manuscript where relevant.

The binding experiments collected with ssDNA may also be measuring affinity to product, or possibly a mix of substrate and product, depending on the activity of SMUG1 under the assay conditions (conc of enzyme, substrate) and timescale of the experiment. This all needs to be resolved so that the experiments are performed under conditions (and timeframe) that enables the substrate to remain intact (U not cleaved) or fully converted to product before the measurements begin.

It would be informative to show binding data for at least one enzyme-DNA combination, in the main text or Supplementary Information.

Lines 356-8. The conclusion that, "This difference in affinity indicates that the process of uracil removal by SMUG1 leads to a stronger binding to the AP site (7 nM) than if SMUG1 binds to a preexisting AP site (66 nM).", should be reconsidered. The product complex generated (almost upon mixing) of SMUG1 and the U or 5FU substrate DNA is a natural AP site with a hydroxyl group at C1'. The complex with "AP site" DNA uses a tetrahydrofuran analog, as indicated by the "idSp" (dSpacer) in the sequence for oligo 5 of Table 1, and this analog lacks the hydroxyl at C1'. The most reasonable interpretation of the results is that the affinity difference is due to the presence of the hydroxyl group for the natural AP DNA product. It is hard to imagine a mechanism whereby "the process of uracil removal could enable stronger binding to the AP site to a preexisting AP site".

On a related note, the main text and legend for Table 1 should state specifically that the "AP site" is a tetrahydrofuran analog.

The authors should consider adding a figure that summarizes the proposed mechanism regarding the orientation of the uracil (5FU) base prior to and after cleavage, including interactions with key active site groups, to clarify their proposed mechanism for SMUG1.

The discussion lacks significant comparison of the current structural and biochemical results to the substantial prior literature on SMUG1, including glycosylase activity for the human enzyme and structural studies by the Verdine and Pearl groups for *Xenopus laevis* SMUG1.

Reviewer #2

(Remarks to the Author)
NCOMMS-25-84477

Structural basis for uracil removal from DNA by human SMUG1
Ludäscher et al

Ludäscher et al describe a structural study of the human SMUG1 protein and its interaction with uracil-containing dsDNA using a combination of techniques including X-ray crystallography, AlphaFold protein structure prediction, neutron crystallography, molecular dynamics. The underlying interest is in the nature of the uracil excision from the DNA and the uracil binding to SMUG1. The paper is well-written, novel, and of high interest for an understanding of the base excision repair process, its significance for cancer biology and for inhibitor design.

The X-ray crystal study of hSMUG1 in complex with a 12bp length of uracil-containing dsDNA is a very nice result – clearly showing the nature of the binding and also showing the glycosidic cleavage of the uracil from the DNA. The authors should mention the rationale for the choice of this particular oligonucleotide sequence for the SMUG1 complex.

In combination with structural studies of SMUG1 with uracil and 5FU, the authors suggest that process that a base-flipping mechanism is involved. They describe MD/FE calculations that suggest that a flipped binding position offers a significant advantage over the native conformation. In addition they have carried out a neutron crystallographic study of deuterium-exchanged SMUG1 complexed to deuterated uracil – this is of particular interest given that it should pin the orientation down definitively. The authors state that the neutron scattering length density maps support the flipped model but the image shown in Supplementary Figure 8B should be improved to bring this out more clearly – perhaps even with a couple of viewing angles rather than a single one that is hard to interpret visually.

Aside from the above points, I feel that the manuscript adds genuinely new insight to the BER process and it suitable for publication in Nature Communications. The work is technically sound and the arguments logically justified using an appropriate array of techniques. The involvement of a neutron crystallographic angle offers a unique scope that could be emphasised more clearly.

Minor points

1. APE1 (appears line 58) acronym should be defined at first use.
2. DKC1 (appears line 385) acronym should be defined at first use.
3. Clarification needed for the footnote in Table 1?

Reviewer #3

(Remarks to the Author)

Overall, this is a great study and well-presented manuscript that significantly advances our structural and mechanistic understanding of human SMUG1. In particular, this is the first report of the human SMUG1 crystal structure and at high resolution. The authors present the first structures of the apo enzyme, product complexes with uracil and 5-fluorouracil, and SMUG1 in complex with double stranded DNA. These are complemented by MD simulations and neutron diffraction analyses that together support a plausible model for the mechanism of SMUG1 base excision. This work could also provide an important foundation for the development of SMUG1 modulators and will have broad impact in the DNA repair and chemical biology communities. This study is of interest to readers of Nature Communications and has the appropriate impact to the field for publication.

My comments below are minor and focused on points of clarity or expansion that would further strengthen the manuscript.

1. It would be helpful for the authors to define more explicitly what is meant by a “productive dsDNA complex.” Does this refer to substrate positioning, or a specific conformational state associated with uracil flipping or excision? This term should probably not be used in the abstract.

2. The abstract is well written but devotes substantial space to broader significance. I recommend adding more detail about the key structural findings and mechanistic insights to better reflect the findings of the study. Some of the broader significance is overstated for the abstract and throughout the document as a whole.

3. Do the presented structures provide any insight into why SMUG1 exhibits distinct substrate specificity (e.g. 5-OHU, 5-HMU) from other members of the UDG superfamily? The manuscript and field would benefit from a brief discussion comparing SMUG1's structural features with those of related enzymes that provide specificity for lesions within the UDG family.

4. The figure related to the neutron scattering density, specifically the statement “The density is weak for the C5 carbon of uracil,” is difficult to interpret as currently presented. The density is not clearly visualized, and the figure does not convincingly support the text. I suggest remaking or clarifying this figure to more directly illustrate the stated observation.

5. The experiment involving SMUG1 and APE1 does not directly demonstrate the mechanism by which hAPE1 outcompetes hSMUG1 from dsDNA following uracil cleavage. Multiple turnover events and the isolation of a product-bound APE1 structure in the crystallization buffer complicate mechanistic interpretation. While the structure itself is interesting and represents a natural AP-site intermediate (which should be published), it does not provide insight into the coordination or competition between these two enzymes at the mechanistic level. Clarifying this point in the discussion would be helpful.

6. The authors note that their results do not align with a previous 2007 study showing that an hSMUG1-R243A mutant retains substantial activity. It would strengthen the manuscript if the authors could expand on this observation and explain the mechanistic basis for the differences between their findings and the earlier study.

Version 1:

Reviewer comments:

Reviewer #1

(Remarks to the Author)

The authors done an excellent job in addressing the concerns raised in the original review. The results provide an important contribution to the field of DNA repair and I think the manuscript is ready for publication.

Reviewer #2

(Remarks to the Author)

Following my original review of this manuscript, the authors have updated the paper to my satisfaction and I now feel that the revised document is suitable for publication.

Reviewer #3

(Remarks to the Author)

The authors have addressed all my concerns. Congrats on a nice study

Dear Editor,

Thank you for the positive response to our manuscript. We also thank the reviewers' for their constructive suggestions. We believe the resulting changes have improved our paper.

We agree with the revisions suggested by the reviewers and have revised our manuscript accordingly. We have addressed these points individually (below) and have uploaded a word document with the changes marked.

REVIEWER COMMENTS

Reviewer #1 (Remarks to the Author):

This manuscript from Stenmark and coworkers reports the first crystal structures of human SMUG1, an important DNA glycosylase that protects against mutations by excising damaged bases including uracil, 5-hydroxyuracil, and other lesions. High resolution structures are presented for the free enzyme, the enzyme bound to uracil and 5-F-uracil, and the enzyme bound to the AP DNA product (without the cleaved U base). Prior structures of SMUG1 were solved for *Xenopus laevis* SMUG1 bound to DNA. The structures and MD simulations of SMUG1 bound to U and 5FU indicate that the base binds in an unexpected orientation, when compared to the prior SMUG1 structures or those of the related enzyme UNG2. suggested the possibility of an interesting and unanticipated mechanism whereby the excised base flips within the active site. The manuscript also reports glycosylase activity and DNA-binding affinity for wild-type SMUG1 and several mutants, though there are issues with the approach and interpretation of these studies. Overall, the results could be of substantial interest to those the field of DNA damage and repair, but the following concerns should be addressed prior to potential publication.

(1) Lines 43-46: the statements “complex networks of DNA repair, also known as DNA damage response (DDR)”, and “the base excision repair (BER) pathway... is the predominant DDR pathway for repairing small base lesions” should be revised. The DDR and DNA repair pathways including BER are distinct processes; BER is not typically considered as a “DDR pathway”

-We have revised the text to remove reference to the DNA damage response (DDR) and instead focus on the role of base excision repair (BER) in repairing small base lesions (page 2).

(2) Line 58: APE1 has endonuclease activity, which is distinct from AP lyase activity

-The text has been changed to read 'endonuclease activity' (page 3).

(3) Line 65: MBD4 is not a member of the UDG superfamily. It belongs to the Endo IV superfamily that also includes MutY. Also, on line 72 “MBD” should be MBD4.

-We thank the reviewer for this correction. As the focus of this work is the UDG superfamily, we have removed MBD4 from this section and revised the text accordingly (page 3).

(4) Lines 104-5: “the structure of SMUG1 in a productive complex with dsDNA”. For an enzyme (such as SMUG1), a “productive complex” typically refers to an enzyme-substrate complex that is poised for the chemical step to occur (which is halted by an enzyme mutation or use of a non-reactive substrate analog). The structure reported here should be referred to as an enzyme-product complex.

-The text has been changed to read “enzyme-product complex of hSMUG1 with dsDNA” (page 4). This has also been updated at several other points within the manuscript (pages 2, 8 and 23).

(5) Lines 151-3: “Upon refinement, it was evident that there was an AP site at the position where uracil was located in the dsDNA sequence, indicating the hSMUG1 successfully cleaved the uracil from the DNA substrate during co-crystallization.” The phrasing sounds as though this result might not have been expected. SMUG1 has very high activity for U excision (kobs ~200 min⁻¹; Stivers group, 2011, Biochemistry 50:618-27) so the AP site is generated as soon as the enzyme and DNA have been mixed, particularly at the concentrations used for crystallization. This should be reworded accordingly, in this section and other sections noting this observation.

-We thank the reviewer for this clarification. We have reworded the text to reflect that AP site formation by hSMUG1 is rapid and expected given its high uracil excision activity, and we have included the Stivers group reference (pages 6-7). The wording has also been revised in the Discussion (page 19).

(6) Fig. 3. The R243 side chain is hard to see in panel B. I may be informative to show more structural detail in panel B, or add another panel with more detail (e.g., stick for AP site and relevant SMUG1 groups, water molecules). It would be helpful to include a simple cartoon figure that shows all of the enzyme-DNA duplex interactions, including phosphate contacts in an easily viewed manner. Fig 3D indicates no contacts with the AP sugar or the phosphate on its 3' side. Is this correct and are there any water-mediated contacts? If so, they should be shown and if not, this should be discussed.

-We thank the reviewer for these helpful suggestions. The side chain of R243 (Figure 3B, page 8) has been modified to improve its visibility. In addition, we have included a new Supplementary Figure (Supplementary Figure 4) that provides more detailed views of the hydrogen-bonding network, including all water-mediated interactions. This figure also includes a simplified cartoon overview showing all amino acids that interact with the dsDNA (Supplementary Figure 4A). We have also highlighted the H-bond interactions at the AP site (Supplementary Figure 4C). The AP sugar is positioned by direct hydrogen bonding with N176 and by a water-mediated hydrogen bond with P86. We also show the H-bond between the 3' end of the AP site and T178, as well as interactions between the 5' phosphate and two serines (S137 and S241). The results text has been updated accordingly to include these details (page 7).

(7) Fig 4. It would be informative to include a figure that shows the alternative (or both) orientations of U in the active site and shows the enzyme interactions, particularly because the orientations cannot be distinguished by electron density.

It would be informative to show H bonds in panel E, perhaps for a representative snapshot from the MD simulations.

*-We thank the reviewer for this suggestion and show hydrogen bonds in the revised version of **Figure 4E** (page 11). We have also updated **Supplementary Figure 6**, which now shows the enzyme interactions in addition to the electron density for both uracil orientations.*

(8) Lines 225-8: The manuscript states that, “While attached to DNA, the uracil can rotate around the N-glycosidic bond. However, the simulation showed that when the base enters the hSMUG1 active site pocket, the uracil is locked in one orientation (“native” pose). Following hydrolysis of the N-glycosidic bond, the free uracil can adopt a “flipped” orientation in the binding pocket due to the symmetry axis along N3 and C6 of the base”. This needs some clarification. Does the simulation indicate the U base is “locked” in the orientation shown in 4E for the modeled enzyme-substrate complex (prior to bond cleavage)? Was a simulation of the ternary product complex performed that shows the U can adopt the orientation in panel 4B without dissociation of the abasic DNA? Or is this conclusion based solely on the structure or simulations of the binary (enzyme-U) complex? This is an important mechanistic question that needs clarification and potentially additional MD studies. It would be helpful to use the terms “enzyme-substrate complex” or “lesion recognition complex” when referring to structures/models of SMUG1 bound to DNA with an intact N-glycosyl bond (rather than “uracil covalently linked”), as these terms are typical in the literature for structures of DNA glycosylases.

*-We thank the reviewer for pointing out that our description of the simulation results was not clear. We have rewritten part of this section (page 10) and also added a reference to **Figure 4E** (page 11), in which the predicted “native” pose of DNA bound to hSMUG1 is shown. We performed MD simulations of both the ternary product complex (**updated Figure 5**, page 13) and the binary enzyme-uracil complex (Free energy calculations in **Supplementary Table 1**). These calculations support that the rotated orientation of uracil is most stable in both the ternary and binary complex. However, our calculations do not reveal if the rotation of the uracil is most likely to occur in the presence or absence of the abasic DNA, which we note in the Discussion (page 22). We agree that “enzyme-substrate complex” is the most appropriate naming of the SMUG1-DNA complex and have introduced the suggested terminology throughout the revised manuscript (pages 9, 22, 31-33).*

(9) Given that nucleotide flipping is mechanism used by DNA glycosylases to extrude the substrate nucleotide out of the DNA duplex, and many readers will associate the terms “flipped” or “flips” with this process. For example, in the abstract, the statement that “the substrate uracil flips following base excision” could confuse many readers. As such, it may be better to use a different term for the potential alternative conformation of the uracil base suggested by the new structures, in this section and throughout the manuscript.

*-We agree with the reviewer and now use the word “rotated” instead “flipped” in the revised manuscript (pages 2, 4, 9-10, 12-13, 21-22, 31 and 33). Please note that the term “rotated” is now also used in **Supplementary Figures 6 and 17**.*

(10) For the MD simulations described on p. 12 of the ternary product complex (SMUG1-DNA-U), was the U found to change its pose during the MD simulation? While the results indicate that the ternary complex with the alternate U pose is more stable, it is important to clarify whether the results provide evidence that the change from the “native” pose (expected just after bond cleavage) and the alternative pose can occur in the ternary product complex, or if it requires dissociation (and rebinding) of the AP DNA.

-In the simulation of the ternary product complex, the MD simulations were initiated from either the “native” or “rotated” poses, and the “rotated” pose shows higher stability (updated Figure 5, page 13). However, our calculations do not reveal if the rotation of the uracil is most likely to occur in the presence or absence of the abasic DNA, which we note in the discussion text (page 22).

(11) Lines 298-307. The finding that SMUG1 binds U tighter than T, based on MD simulations is interesting, but additional simulations modeling DNA-bound SMUG1 with an intact dT substrate flipped into the active site would be much more informative and could be compared to that shown in Fig 4E for dU.

-We thank the reviewer for this excellent suggestion and have performed the requested simulations of DNA-bound SMUG1 with an intact dT substrate. We then compared these new calculations to the simulations with a uracil substrate. Whereas the interactions with key residues M84, F98, N163 and H239 were maintained in the simulations with the uracil substrate, introduction of the thymine led to displacement of the base and none of the five hydrogen bonds were formed. For example, uracil formed stable hydrogen bonds with N163, with an occupancy of 80%, whereas the same interaction is observed in less than 10% of the simulations with thymine (Supplementary Figure 10). These calculations hence support that thymine does not fit as well as uracil in the bindings site, in agreement with experimental data. We have added these new data to the results section of the revised manuscript (pages 14 and 22).

(12) Lines 321-3. Regarding the structure of DNA-bound APE1 resulting from a crystallization sample that contained a SMUG1 substrate and both SMUG1 and APE1, this is an end state that could be reached by many potential mechanisms. I think the only conclusion to be drawn is that under the conditions used for crystallization (very high concentrations of both enzymes and DNA), that APE1 has higher affinity for its substrate (AP DNA) than does SMUG1 for its product. I think the statement that, “the absence of hSMUG1 shows that hAPE1 releases hSMUG1 from the AP site containing dsDNA by outcompeting hSMUG1 prior to backbone cleavage by hAPE1, rather than by cleaving the DNA backbone first in order to facilitate hSMUG1 release”, should be revised to avoid the suggestion of an active mechanism by APE1. It could be active, but this can’t be ascertained by the structure. Also, given that APE1 acts by flipping the AP nucleotide into its active site, it seems impossible that it might be able to do so when SMUG1 is bound to the AP site. I think the latter part of this sentence should be revised because it isn’t really a plausible mechanism given what is known about APE1.

-We thank the reviewer for this suggestion. We agree that the structure represents a final, product-bound state and does not directly reveal the mechanism of hSMUG1 release. We have revised the results text (page 15) and discussion text (page 21) to clarify that the absence of hSMUG1 in the crystal structure reflects the higher apparent affinity of hAPE1 for AP sites under the crystallization conditions, without implying an active release mechanism. We also note that simultaneous cleavage by hAPE1 while SMUG1 is bound is unlikely, because AP nucleotide flipping into the APE1 active site is required for catalysis.

(13) Regarding the APE1-AP-DNA structure, which mimics the enzyme-substrate complex, it would be informative to briefly discuss how this structure advances, if at all, upon prior structural and mechanistic results for APE1.

-We thank the reviewer for this helpful suggestion. We now explicitly reference prior human APE1-dsDNA structures (DOI: 10.1038/nsmb.3105; 10.1038/35000249; 10.1074/jbc.M112.422774) and compare them with our complex in the revised manuscript (page 15). Overall, our structure is very similar to previously described APE1-dsDNA complexes, while uniquely containing an abasic sugar with a C1' hydroxyl group instead of the commonly used THF analog, providing a closer chemical approximation to a native AP site. However, we also clarify in the text that this DNA substrate is still not fully physiological due to its phosphothioate modifications flanking the AP-site. We have updated **Supplementary Figure 11** to include a comparison with PDB entry 1DEW (<https://doi.org/10.1038/35000249>).

(14) Fig 6B. The activity data for w/t and mutant SMUG1 is shown and apparently fitted to the MM equation but the resulting parameters (k_{cat} , K_m , and k_{cat}/K_m) are not provided. These need to be included in a table and discussed.

-We thank the referee for the comments regarding figure 6B. We apologize for not including this table previously. A table with the calculated parameters (k_{cat} , K_m , and k_{cat}/K_m) is now included as **Supplementary Table 5** (referenced in main text on page 16).

(15) Fig 6B (cont.). The duration of the experiments appears to be too short by ~twofold for wild-type enzyme and much too short for the mutants to give a reliable fit to the MM eqn; this is a hyperbolic eqn and the experiments need to be collected at a high enough substrate concentration to saturate the enzyme. This is not the case for the wild-type enzyme and cannot be readily judged for the mutants because the y-axis scale is too large. The mutant results should be shown in a different panel with a y-axis that is small enough in range to see whether the substrate concentration is high enough to give a reliable fitting for V_{max} .

-We thank the referee for this comment. We have now separated the saturation curves of the wt enzyme from those of the mutants (now added as **Supplementary Figure 13**, referenced in main text on page 16). As the referee correctly points out, neither the wt nor the mutant is saturated at the highest concentration of substrate used. The aim here is not to provide a correct turnover number but to show the difference in activity between the mutants and wildtype SMUG1. However, we have also fitted an equation from which the parameter k_{cat}/K_m (the initial slope of the Michaelis Menten curve) is obtained directly (and not as the quotient of k_{cat} and K_m

obtained by fitting the MM equation to the data) and provides a proper comparison of the catalytic efficiencies of the SMUG1 wt and mutants.

(16) Fig 6B (cont.) The main text should state that the enzymatic assay is a coupled assay, where the fluorescence signal is produced by a second enzyme, APE1, which acts on the product (AP DNA) generated by SMUG1. Also, controls are needed to demonstrate that the reactions are collected under conditions such that the observed activity is dependent entirely on the activity of SMUG1 rather than APE1. The activity of SMUG1 is relatively high and the observed change in fluorescence could be due in part to APE1.

-We have now added to the manuscript text that the assay is a coupled assay (pages 16 and 28). During assay development we titrated APE1 to make sure that the concentration used is not rate limiting and that the signal is dependent on SMUG1 activity only. An excess of APE1 corresponding to 20-fold the amount required to cleave 100 nM substrate was used. Controls without enzymes or with APE1 only were included in the experiment. Added to the Supplementary is now a figure (**Supplementary Figure 12B**, referenced on page 16) showing the fluorescence signal after full conversion of the substrate at 0, 10, 20, 30, 40 and 60 nM substrate with SMUG1+APE1, APE1 only and without enzymes, showing that the contribution of APE1 to the increase in fluorescence (and formation of product in absence of SMUG1) is negligible and similar to the signal obtained in absence of both SMUG1 and APE1. Moreover, when calculating the initial rates at different substrate concentrations the fluorescence from wells with APE1 only was subtracted from those with SMUG1 and APE1.

(17) Fig 6B (cont.) Ideally, the units for the y-axis should be converted to product concentration per min rather than fluorescence change per min. This would allow the results to be compared quantitatively to previous studies of human SMUG1, which should be included in the manuscript.

- A standard curve based on Δ Fluorescence after complete conversion of the substrate, using 10 nM SMUG1 and different concentrations of substrate, was used to convert the units for the y-axis to reaction rate (v) (**Supplementary Figure 12B**, referred to in Methods section, page 29).

Figure 6B has also been updated (page 17) to show the requested y-axis values.

(18) Fig 6B (cont.). It would be informative to show the raw data from the fluorescence assay for at least one enzyme-DNA combination, including for all substrate concentrations used for fitting the results in Fig 6B, in the main text or Supplementary Information.

-We thank the reviewer for this suggestion. The graphs with raw data from which the initial reaction rates for the wildtype enzyme were calculated are now shown in **Supplementary Figure 19** (referenced in Methods section, page 29).

(19) The DNA binding experiments and results need some clarification. Given the very high activity of SMUG1 for U and 5FU, the binding assay for wild-type enzyme is likely measuring SMUG1 affinity to its product(s), depending on the concentration

of enzyme and DNA substrate. Regarding the latter, the Methods section indicates a 5 mM final concentration of DNA was used, which seems much too high. Perhaps it was 5 nM? Controls are needed to assess the extent to which DNA substrates are converted to AP DNA product during the timeframe of the experiment, for w/t and mutant enzymes. These details should be clarified at the beginning of this topic in the results section (line ~350) and in Table 1, and throughout the manuscript where relevant.

-We thank the reviewer for highlighting this point. The DNA concentration reported in the Methods section (5 mM) was a typo; the actual concentration used was 5 nM (page 28). To clarify the extent to which binding reflects substrate versus product, we performed additional MST experiments using the SMUG1 N85A mutant, which is catalytically inactive but does not alter the uracil-binding pocket (<https://doi.org/10.1093/nar/gkh859> & <https://doi.org/10.1093/nar/gkm372>). The final result of similar values for wild-type SMUG1 and SMUG1-N85A does not allow us to assign the affinity values to for the wildtype SMUG1-oligo complex to either the educt or product bound complex and hence we have to assume that the measurement is a mixture. Therefore statements regarding binding affinities in the manuscript have been changed to “apparent” affinities (pages 17 and 20) to clarify that the MST values represent the total experimental environment (e.g., educt, product and off-target binding). An appropriate statement has been included in the manuscript at the beginning of the section for the MST measurements (page 17).

(20) The binding experiments collected with ssDNA may also be measuring affinity to product, or possibly a mix of substrate and product, depending on the activity of SMUG1 under the assay conditions (conc of enzyme, substrate) and timescale of the experiment. This all needs to be resolved so that the experiments are performed under conditions (and timeframe) that enables the substrate to remain intact (U not cleaved) or fully converted to product before the measurements begin.

*-We agree with the reviewer that binding experiments with ssDNA may measure a mixture of substrate- and product-bound complexes, depending on the activity of SMUG1 under the assay conditions. As described in **point #19** above, MST measurements were performed using both wildtype SMUG1 and the N85A inactive mutant, allowing assessment of substrate binding (page 18). The resulting similarity in affinities between wildtype and N85A does not allow us to assign the affinity values for the wildtype SMUG1-oligo complex to either the substrate or product bound complex and hence we have to assume that the measurement is a mixture. Accordingly, all reported affinities have been changed to “apparent” affinities in the manuscript (pages 17 and 20).*

(21) It would be informative to show binding data for at least one enzyme-DNA commination, in the main text or Supplementary Information.

*-A new figure (**Supplementary Figure 14**) has now been added which shows the requested binding data for wt SMUG1, SMUG1-R124A, SMUG1-R215A and SMUG1-R243A (referenced in manuscript on page 17).*

(22) Lines 356-8. The conclusion that, “This difference in affinity indicates that the process of uracil removal by SMUG1 leads to a stronger binding to the AP site (7

nM) than if SMUG1 binds to a preexisting AP site (66 nM).”, should be reconsidered. The product complex generated (almost upon mixing) of SMUG1 and the U or 5FU substrate DNA is a natural AP site with a hydroxyl group at C1'. The complex with “AP site” DNA uses a tetrahydrofuran analog, as indicated by the “/idSp/” (dSpacer) in the sequence for oligo 5 of Table 1, and this analog lacks the hydroxyl at C1'. The most reasonable interpretation of the results is that the affinity difference is due to the presence of the hydroxyl group for the natural AP DNA product. It is hard to imagine a mechanism whereby “the process of uracil removal could enable stronger binding to the AP site to a preexisting AP site”.

*-We agree with the reviewer and have altered the previous statement. The text (page 17) now reads: “The low-nanomolar apparent affinity observed with uracil-containing dsDNA likely reflects binding under catalytic conditions, whereas the weaker affinity for AP-site dsDNA may partly result from using a tetrahydrofuran abasic-site analog, which differs chemically from a natural AP site (**Supplementary Figure 15**).*

(23) On a related note, the main text and legend for Table 1 should state specifically that the “AP site” is a tetrahydrofuran analog.

*-**Table 1** and the manuscript text now states that the AP-site dsDNA used in this study is a tetrahydrofuran analog (pages 17 and 18). We have also included a new figure (**Supplementary Figure 15**), which shows how this analog differs to a naturally occurring AP-site.*

(24) The authors should consider adding a figure that summarizes the proposed mechanism regarding the orientation of the uracil (5FU) base prior to and after cleavage, including interactions with key active site groups, to clarify their proposed mechanism for SMUG1.

*-We thank the reviewer for this helpful suggestion. We agree that such a schematic is valuable and have added a figure summarizing the proposed mechanism for uracil orientation before and after cleavage, including key active-site interactions (**Supplementary Figure 16**), which is now referenced in the Discussion (page 22).*

(25) The discussion lacks significant comparison of the current structural and biochemical results to the substantial prior literature on SMUG1, including glycosylase activity for the human enzyme and structural studies by the Verdine and Pearl groups for *Xenopus laevis* SMUG1.

*-We thank the reviewer for this helpful comment. We have expanded the Supplementary Information to include an additional structural comparison with *Xenopus laevis* SMUG1 (**Supplementary Figure 5C-D**), demonstrating the strong conservation of the uracil-binding pocket (page 9). This analysis complements our existing comparisons with DNA-bound xSMUG1 structures. We also note that our DNA-bound structure provides direct structural evidence for the DNA-penetrating loop proposed previously by the Verdine and Pearl (page 19).*

In addition, we now compare our biochemical data with earlier kinetic analyses of SMUG1 (page 20). Kinetic parameters determined for SMUG1 and reported by Matsubara et al. for the wildtype enzyme doi: [10.1093/nar/gkh859](https://doi.org/10.1093/nar/gkh859) were k_{cat} (min⁻¹)=

0.84, $K_m = 2.2$ (nM), $k_{cat}/K_m = 0.38$ ($\text{min}^{-1}\text{nM}^{-1}$) and k_{obs} (k_{cat}) was reported in doi: [10.3390/molecules24173133](https://doi.org/10.3390/molecules24173133) to be 0.4 min^{-1} (0.0067 s^{-1}). We do, however, observe a higher K_m than previously reported. The reason for this may be due to the use of the fluorophore and quencher-labelled substrate used in our study, not binding with the same affinity to the SMUG1 enzyme as the radioactively labelled substrates used in the above-mentioned publications.

Reviewer #2 (Remarks to the Author):

NCOMMS-25-84477

Structural basis for uracil removal from DNA by human SMUG1
Ludäscher et al

Ludäscher et al describe a structural study of the human SMUG1 protein and its interaction with uracil-containing dsDNA using a combination of techniques including X-ray crystallography, AlphaFold protein structure prediction, neutron crystallography, molecular dynamics. The underlying interest is in the nature of the uracil excision from the DNA and the uracil binding to SMUG1. The paper is well-written, novel, and of high interest for an understanding of the base excision repair process, its significance for cancer biology and for inhibitor design.

(1) The X-ray crystal study of hSMUG1 in complex with a 12bp length of uracil-containing dsDNA is a very nice result – clearly showing the nature of the binding and also showing the glycosidic cleavage of the uracil from the DNA. The authors should mention the rationale for the choice of this particular oligonucleotide sequence for the SMUG1 complex.

- The oligonucleotide was chosen based on previous structural work on Xenopus SMUG1–DNA complexes, which interestingly, did not capture either the enzyme-substrate or enzyme-product complex. The 12 bp length provides a stable duplex while remaining well suited for crystallization, as longer DNA can introduce flexibility and hinder lattice formation. The manuscript text has been updated accordingly (page 6).

(2) In combination with structural studies of SMUG1 with uracil and 5FU, the authors suggest that process that a base-flipping mechanism is involved. They describe MD/FE calculations that suggest that a flipped binding position offers a significant advantage over the native conformation. In addition they have carried out a neutron crystallographic study of deuterium-exchanged SMUG1 complexed to deuterated uracil – this is of particular interest given that it should pin the orientation down definitively. The authors state that the neutron scattering length density maps support the flipped model but the image shown in Supplementary Figure 8B should be improved to bring this out more clearly – perhaps even with a couple of viewing angles rather than a single one that is hard to interpret visually.

*-We thank the reviewer for this suggestion. We have revised **Supplementary Figure 9** to include another panel (**panel C**) which more clearly shows the neutron scattering density for the uracil base (referenced in the main text on page 13).*

(3) Aside from the above points, I feel that the manuscript adds genuinely new insight to the BER process and it suitable for publication in Nature Communications. The work is technically sound and the arguments logically justified using an appropriate array of techniques. The involvement of a neutron crystallographic angle offers a unique scope that could be emphasised more clearly.

We thank the reviewer for their positive and encouraging assessment of the manuscript and for highlighting the value of the neutron crystallographic approach. We have revised the manuscript to more clearly emphasise the uniqueness and significance of this methodology. A survey of the Protein Data Bank indicates that only 165 joint neutron/X-ray structures have been deposited to date, of which just 35 correspond to human proteins. To our knowledge, the present study represents the first neutron structure of a DNA-binding protein, and specifically of a DNA glycosylase involved in base excision repair. We have updated the Results section accordingly (page 14).

Minor points:

(1) APE1 (appears line 58) acronym should be defined at first use.

-The acronym APE1 is now defined at first use (page 3).

(2) DKC1 (appears line 385) acronym should be defined at first use.

-The acronym DKC1 is now defined at first use (page 19).

(3) Clarification needed for the footnote in Table 1?

*-The footnote in **Table 1** (page 18) has been revised for clarity. It now explicitly defines "NB" as no detectable binding under the conditions tested and clarifies that "WT" refers to the N-terminally truncated hSMUG1-Δ24 construct used throughout the study for crystallography, enzymatic assays, and MST measurements.*

Reviewer #3 (Remarks to the Author):

Overall, this is a great study and well-presented manuscript that significantly advances our structural and mechanistic understanding of human SMUG1. In particular, this is the first report of the human SMUG1 crystal structure and at high resolution. The authors present the first structures of the apo enzyme, product complexes with uracil and 5-fluorouracil, and SMUG1 in complex with double stranded DNA. These are complemented by MD simulations and neutron diffraction analyses that together support a plausible model for the mechanism of SMUG1 base excision. This work could also provide an important foundation for the development of SMUG1 modulators and will have broad impact in the DNA repair and chemical biology communities. This study is of interest to readers of Nature Communications and has the appropriate impact to the field for publication.

My comments below are minor and focused on points of clarity or expansion that would further strengthen the manuscript.

(1) It would be helpful for the authors to define more explicitly what is meant by a “productive dsDNA complex.” Does this refer to substrate positioning, or a specific conformational state associated with uracil flipping or excision? This term should probably not be used in the abstract.

*-We agree that the term “productive dsDNA complex” was ambiguous and could be misleading. As also addressed in response to **Reviewer 1, Comment #4**, we have replaced this terminology throughout the manuscript (pages 4 and 23) with “enzyme-product complex,” which more accurately reflects the nature of the structure. The term has also been removed from the abstract (page 2).*

(2) The abstract is well written but devotes substantial space to broader significance. I recommend adding more detail about the key structural findings and mechanistic insights to better reflect the findings of the study. Some of the broader significance is overstated for the abstract and throughout the document as a whole.

-We have reworded the abstract to better highlight the key findings of the study (page 2). Regarding the broader significance of our work, we respectfully disagree with the reviewer. A detailed biochemical, biophysical, and structural understanding of the target in question is important. Establishing a reproducible, high-resolution, structural biology platform for a human drug target is of fundamental importance for drug discovery and development. In this study, we present a mature and robust platform that can be readily utilized by both academia and industry to facilitate and accelerate drug development programs. We and others have demonstrated the value of such platforms across multiple previous projects, in which similar approaches have directly enabled target validation (or devalidation), structure-guided drug design, and therapeutic development.

Representative example from our team include:

*Luttens A et al. *Nat Commun.* 2025;16:1741.*

*Michel M et al. *Science.* 2022;376:1471–1476.*

*Zhang SM et al. *Nat Chem Biol.* 2020;16:1120–1128.*

*Visnes T et al. *Science.* 2018;362:834–839.*

*Rudling A et al. *J Med Chem.* 2017;60:8160–8169.*

*Gustafsson R et al. *Cancer Res.* 2017;77:937–948.*

*Gad H et al. *Nature.* 2014;508:215–221.*

(3) Do the presented structures provide any insight into why SMUG1 exhibits distinct substrate specificity (e.g. 5-OHU, 5-HMU) from other members of the UDG superfamily? The manuscript and field would benefit from a brief discussion comparing SMUG1’s structural features with those of related enzymes that provide specificity for lesions within the UDG family.

*-We thank the reviewer for this helpful suggestion. To better place our findings in the context of the UDG superfamily, we have added a new supporting figure (**Supplementary Figure 18**) that directly compares uracil recognition in SMUG1 with the corresponding binding pockets of UNG2 and TDG. We have also expanded the*

Discussion (page 23) to summarize the similarities and differences in uracil recognition, and how this may influence substrate selectivity.

(4) The figure related to the neutron scattering density, specifically the statement “The density is weak for the C5 carbon of uracil,” is difficult to interpret as currently presented. The density is not clearly visualized, and the figure does not convincingly support the text. I suggest remaking or clarifying this figure to more directly illustrate the stated observation.

*-We thank the reviewer for noting that the neutron scattering density for the C5 carbon of uracil was difficult to interpret in the original figure. We agree that this was not presented as clearly as it could have been. To address this, we have added an additional panel (**Supplementary Figure 9C**) that more clearly shows the neutron scattering density associated with the uracil base, making the weak density at the C5 position easier to evaluate (referenced in manuscript text on page 13).*

(5) The experiment involving SMUG1 and APE1 does not directly demonstrate the mechanism by which hAPE1 outcompetes hSMUG1 from dsDNA following uracil cleavage. Multiple turnover events and the isolation of a product-bound APE1 structure in the crystallization buffer complicate mechanistic interpretation. While the structure itself is interesting and represents a natural AP-site intermediate (which should be published), it does not provide insight into the coordination or competition between these two enzymes at the mechanistic level. Clarifying this point in the discussion would be helpful.

-We thank the reviewer for this comment. We have clarified in the results (page 15) and discussion (page 21) that the crystal structure represents a product-bound intermediate and does not directly reveal the mechanistic coordination or competition between hAPE1 and hSMUG1. Our revised text emphasizes that the observed absence of hSMUG1 likely reflects the higher apparent affinity of hAPE1 for AP-sites under crystallization conditions, rather than an active displacement mechanism. Additionally, we note that simultaneous DNA backbone cleavage by hAPE1 while SMUG1 is bound is unlikely due to the requirement for AP nucleotide flipping into the APE1 active site.

(6) The authors note that their results do not align with a previous 2007 study showing that an hSMUG1-R243A mutant retains substantial activity. It would strengthen the manuscript if the authors could expand on this observation and explain the mechanistic basis for the differences between their findings and the earlier study.

-As the reviewer correctly notes, our results for the hSMUG1-R243A mutant differ significantly from those reported in an earlier study by Pettersen et al. (doi:10.1093/nar/gkm372). This discrepancy may result from differences in how the activity was measured. Specifically, the activity of hSMUG1 R243A in the publication from 2007 was measured at a single substrate concentration of 20 nM using a radioactively labelled substrate compared to in this study where we use a fluorophore and quencher labelled substrate and monitor the activity at concentrations ranging from 0 to 160 nM. The addition of a fluorophore and quencher on the substrate is likely to reduce substrate affinity and hence increase

K_m. The enzyme is probably saturated at 20 nM of the radioactively labelled substrate based on the reported *K_m* of 2 nM for SMUG1 wt (doi: [10.1093/nar/gkh859](https://doi.org/10.1093/nar/gkh859)) and differences in activity between the R243A and wt at lower nonsaturating substrate concentrations would not be observed under these conditions. We have updated the discussion text accordingly (page 20).

Dear Editor,

Thank you for the positive response to our manuscript. We are grateful to the reviewers for their constructive feedback and are pleased that our revisions have satisfactorily addressed their comments. We believe their suggestions have helped to strengthen the manuscript significantly.

REVIEWER COMMENTS

Reviewer #1 (Remarks to the Author):

The authors done an excellent job in addressing the concerns raised in the original review. The results provide an important contribution to the field of DNA repair and I think the manuscript is ready for publication.

Reviewer #2 (Remarks to the Author):

Following my original review of this manuscript, the authors have updated the paper to my satisfaction and I now feel that the revised document is suitable for publication.

Reviewer #3 (Remarks to the Author):

The authors have addressed all my concerns. Congrats on a nice study